# Enhancing Semi-Supervised Learning via Representative and Diverse Sample Selection

**Qian Shao[1,3]\***, **Jiangrui Kang[2]\***, **Qiyuan Chen[1,3]\***, **Zepeng Li[4]**, **Hongxia Xu[1,3]**,
**Yiwen Cao[2]**, **Jiajuan Liang[2]†**, **and Jian Wu[1]†**

[1]College of Computer Science & Technology and Liangzhu Laboratory, Zhejiang University
[2]BNU-HKBU United International College  [3]WeDoctor Cloud
[4]The State Key Laboratory of Blockchain and Data Security, Zhejiang University
{qianshao, qiyuanchen, lizepeng, einstein, wujian2000}@zju.edu.cn
{kangjiangrui, yiwencao, jiajuanliang}@uic.edu.cn

## Abstract

Semi-Supervised Learning (SSL) has become a preferred paradigm in many deep learning tasks, which reduces the need for human labor. Previous studies primarily focus on effectively utilising the labelled and unlabeled data to improve performance. However, we observe that how to select samples for labelling also significantly impacts performance, particularly under extremely low-budget settings. The sample selection task in SSL has been under-explored for a long time. To fill in this gap, we propose a Representative and Diverse Sample Selection approach (RDSS). By adopting a modified Frank-Wolfe algorithm to minimise a novel criterion $\alpha$-Maximum Mean Discrepancy ($\alpha$-MMD), RDSS samples a representative and diverse subset for annotation from the unlabeled data. We demonstrate that minimizing $\alpha$-MMD enhances the generalization ability of low-budget learning. Experimental results show that RDSS consistently improves the performance of several popular SSL frameworks and outperforms the state-of-the-art sample selection approaches used in Active Learning (AL) and Semi-Supervised Active Learning (SSAL), even with constrained annotation budgets. Our code is available at RDSS.

## 1 Introduction

Semi-Supervised Learning (SSL) is a popular paradigm which reduces reliance on large amounts of labeled data in many deep learning tasks [40, 59]. Previous SSL research mainly focuses on effectively utilising labelled and unlabeled data. Specifically, labelled data directly supervise model learning, while unlabeled data help learn a desirable model that makes consistent and unambiguous predictions [53]. Besides, we also find that how to select samples for annotation will greatly affect model performance, particularly under extremely low-budget settings (see Section 7.2).

The prevailing sample selection methods in SSL have many shortcomings. For example, random sampling may introduce imbalanced class distributions and inadequate coverage of the overall data distribution, resulting in poor performance. Stratified sampling randomly selects samples within each class, which is impractical in real-world scenarios where the label for each sample is unknown. Existing researchers also employ representativeness and diversity strategies to select appropriate samples for annotation. Representativeness [13] ensures that the selected subset distributes similarly with the entire dataset, and diversity [54] is designed to select informative samples by pushing away

---

\*These authors contributed equally to this work.
†Corresponding authors.

38th Conference on Neural Information Processing Systems (NeurIPS 2024).

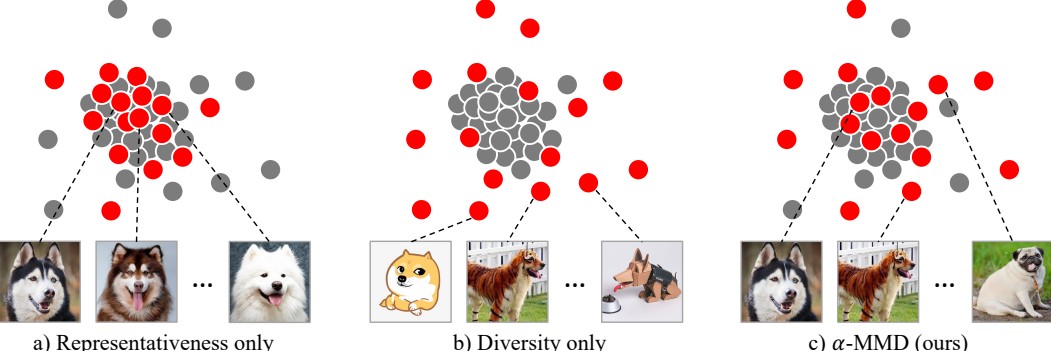

| a) Representativeness only | b) Diversity only | c) $\alpha$-MMD (ours) |

Figure 1: Visualization of selected samples from a dog dataset. The red and grey circles respectively symbolize the selected and unselected samples. **a)** The selected samples often contain an excessive number of highly similar instances, leading to redundancy; **b)** The selected samples contain too many edge points, unable to cover the entire dataset; **c)** The selected samples represent the entire dataset comprehensively and accurately.

them in feature space. And focusing on only one aspect presents significant limitations (Figure 1a and b). To address these issues, Xie et al. [57] and Wang et al. [50] employ a combination of the two strategies for sample selection. These methods set a fixed ratio for representativeness and diversity, restricting the ultimate performance through our empirical evidence (see Section 7.4). Fundamentally, they lack a theoretical basis to substantiate their effectiveness.

We observe that Active Learning (AL) primarily focuses on selecting the right samples for annotation, and numerous studies transfer the sample selection methods of AL into SSL, giving rise to Semi-Supervised Active Learning (SSAL) [51]. However, most of these approaches exhibit several limitations: (1) They require randomly selected samples to begin with, which expends a portion of the labelling budget, making it difficult to work effectively with a very limited budget (*e.g.*, 1% or even lower) [6]; (2) They involve human annotators in iterative cycles of labelling and training, leading to substantial labelling overhead [57]; (3) They are coupled with the model training so that samples for annotation need to be re-selected every time a model is trained [50]. In summary, selecting the appropriate samples for annotation is challenging in SSL.

To address these challenges, we propose a Representative and Diverse Sample Selection approach (RDSS) that requests annotations only once and operates independently of the downstream tasks. Specifically, inspired by the concept of Maximum Mean Discrepancy (MMD) [14], we design a novel criterion named $\alpha$-MMD. It aims to strike a balance between representativeness and diversity via a trade-off parameter $\alpha$ (Figure 1c), for which we find an optimal interval adapt to different budgets. By using a modified Frank-Wolfe algorithm called Generalized Kernel Herding without Replacement (GKHR), we can get an efficient approximate solution to this minimization problem.

We prove that under certain Reproducing Kernel Hilbert Space (RKHS) assumptions, $\alpha$-MMD effectively bounds the difference between training with a constrained versus an unlimited labelling budget. This implies that our proposed method could significantly enhance the generalization ability of learning with limited labels. We also give a theoretical assessment of GKHR with some supplementary numerical experiments, showing that GKHR performs well in learning with limited labels.

Furthermore, we evaluate our proposed RDSS across several popular SSL frameworks on the datasets CIFAR-10/100 [19], SVHN [30], STL-10 [9] and ImageNet [10]. Extensive experiments show that RDSS outperforms other sample selection methods widely used in SSL, AL or SSAL, especially with a constrained annotation budget. Besides, ablation experimental results demonstrate that RDSS outperforms methods using a fixed ratio.

The main contributions of this article are as follows:

- We propose RDSS, which selects representative and diverse samples for annotation to enhance SSL by minimizing a novel criterion $\alpha$-MMD. Under low-budget settings, we develop a fast and efficient algorithm, GKHR, for optimization.

- We prove that our method benefits the generalizability of the trained model under certain assumptions and rigorously establish an optimal interval for the trade-off parameter $\alpha$ adapt to the different budgets.
- We compare RDSS with sample selection strategies widely used in SSL, AL or SSAL, the results of which demonstrate superior sample efficiency compared to these strategies. In addition, we conduct ablation experiments to verify our method's superiority over the fixed-ratio approach.

## 2   Related Work

**Semi-Supervised Learning**   Semi-Supervised Learning (SSL) effectively utilizes sparse labeled data and abundant unlabeled data for model training. Consistency Regularization [34, 20, 45], Pseudo-Labeling [21, 56] and their hybrid strategies [40, 63, 35] are commonly used in SSL. Consistency Regularization ensures the model's output stays stable even when there's noise or small changes in the input, usually from the data augmentation [55]. Pseudo-labelling integrates high-confidence data pseudo-labels directly into training, adhering to entropy minimization [23]. Moreover, an integrative approach that combines the aforementioned strategies can also achieve substantial results [53, 59]. Even though these approaches have been proven effective, they usually assume that labelled samples are randomly selected from each class (*i.e.,* stratified sampling), which is not practical in real-world scenarios where the label for each sample is unknown.

**Active Learning**   Active learning (AL) aims to optimize the learning process by selecting the appropriate samples for labelling, reducing reliance on large labelled datasets. There are two different criteria for sample selection: uncertainty and representativeness. Uncertainty sampling selects samples about which the current model is most uncertain. Earlier studies utilized posterior probability [22, 49], entropy [18, 26], and classification margin [47] to estimate uncertainty. Recent research regards uncertainty as training loss [17, 60], influence on model performance [11, 24] or the prediction discrepancies between multiple classifiers [8]. However, uncertainty sampling methods may exhibit performance disparities across different models, leading researchers to focus on representativeness sampling, which aims to align the distribution of selected subset with that of the entire dataset [36, 39, 27]. Most AL approaches are difficult to perform well under extremely low-label settings. This may be because they usually require randomly selected samples to begin with and involve human annotators in iterative cycles of labelling and training, leading to substantial labelling overhead.

**Model-Free Subsampling**   Subsampling is a statistical approach which selects a subset with size $m$ as a surrogate for the full dataset with size $n \gg m$. While model-based subsampling methods depend heavily on the model assumptions [1, 61], improper choice of the model could lead to bad performance of estimation and prediction. In that case, model-free subsampling is preferred in data-driven modelling tasks, as it does not depend on the model assumptions. There are mainly two kinds of popular model-free subsampling methods. The one is induced by minimizing statistical discrepancies, which forces the distribution of subset to be similar to that of full data, in other words, selects representative subsamples, such as Wasserstein distance [13], energy distance [28], uniform design [65], maximum mean discrepancy [7] and generalized empirical $F$-discrepancy [66]. The other tends to select a diverse subset containing as many informative samples as possible [54]. The above-mentioned methodologies either exclusively focus on representativeness or diversity, which are difficult to effectively apply to SSL.

## 3   Problem Setup

Let $\mathcal{X}$ be the unlabeled data space, $\mathcal{Y}$ be the label space, $\mathbf{X}_n = \{\mathbf{x}_i\}_{i \in [n]} \subset \mathcal{X}$ be the full unlabeled dataset containing pairwise different data, and $\mathcal{I}_m = \{i_1, i_2, \cdots, i_m\} \subset [n](m < n)$ be an index set contained in $[n]$, our goal is to find an index set $\mathcal{I}_m^* = \{i_1^*, i_2^*, \cdots, i_m^*\} \subset [n](m < n)$ such that the selected set of samples $\mathbf{X}_{\mathcal{I}_m^*} = \{\mathbf{x}_{i_1^*}, \mathbf{x}_{i_2^*}, \cdots, \mathbf{x}_{i_m^*}\}$ is the most informative. After that, we can get access to the true labels of selected samples and use the set of labelled data $S = \{(\mathbf{x}_i, y_i)\}_{i \in \mathcal{I}_m^*}$ and the rest of the unlabeled data to train a deep learning model.

Following the methodology of previous works, we use representativeness and diversity as criteria for evaluating the informativeness of selected samples. Representativeness ensures the selected samples distribute similarly to the full unlabeled dataset. Diversity is proposed to prevent an excessive concentration of selected samples in high-density areas of the full unlabeled dataset. Furthermore, the cluster assumption in SSL suggests that the data tend to form discrete clusters, in which boundary points are likely to be located in the low-density area. Therefore, under this assumption, selected samples with diversity contain more boundary points than the non-diversified ones, which is desired in training classifiers.

As a result, our goal can be formulated by solving the following problem:

$$\max_{\mathcal{I}_m \subset [n]} \mathrm{Rep}(\mathbf{X}_{\mathcal{I}_m}, \mathbf{X}_n) + \lambda \mathrm{Div}(\mathbf{X}_{\mathcal{I}_m}, \mathbf{X}_n), \tag{1}$$

where $\mathrm{Rep}(\mathbf{X}_{\mathcal{I}_m}, \mathbf{X}_n)$ and $\mathrm{Div}(\mathbf{X}_{\mathcal{I}_m}, \mathbf{X}_n)$ quantify the representativeness and diversity of selected samples respectively and $\lambda$ is a hyperparameter to balance the trade-off representativeness and diversity.

Besides, we propose another two fundamental settings which are beneficial to the implementation of the framework: **(1) Low-budget learning.** The budget for many of the real-world tasks which require sample selection procedures is relatively low compared to the size of unlabeled data. Therefore, we set $m/n \leq 0.2$ in default in the following context, including the analysis of the sampling algorithm and the experiments; **(2) Sampling without Replacement.** Compared with the setting of sampling with replacement, sampling without replacement offers several benefits which better match our tasks, including bias and variance reduction, precision increase and representativeness enhancement [25, 46].

## 4 Representative and Diversity Sample Selection

The Representative and Diverse Sample Selection (RDSS) framework consists of two steps: (1) *Quantification.* We quantify the representativeness and diversity of selected samples by a novel concept called $\alpha$-MMD (6), where $\lambda$ is replaced by $\alpha$ as the trade-off hyperparameter; (2) *Optimization.* We optimize $\alpha$-MMD by GKHR algorithm to obtain the optimally selected samples $\mathbf{X}_{\mathcal{I}_m^*}$.

### 4.1 Quantification of Diversity and Representativeness

In classical statistics and machine learning problems, the inner product of data points $\mathbf{x}, \mathbf{y} \in \mathcal{X}$, defined by $\langle \mathbf{x}, \mathbf{y} \rangle$, is employed to as a similarity measure between $\mathbf{x}, \mathbf{y}$. However, the application of linear functions can be very restrictive in real-world problems. In contrast, kernel methods use kernel functions $k(\mathbf{x}, \mathbf{y})$, including Gaussian kernels (RBF), Laplacian kernels and polynomial kernels, as non-linear similarity measures between $\mathbf{x}, \mathbf{y}$, which are actually inner products of the projections of $k(\mathbf{x}, \mathbf{y})$ in some high-dimensional feature space [29].

Let $k(\cdot, \cdot)$ be a kernel function on $\mathcal{X} \times \mathcal{X}$, and we employ $k(\cdot, \cdot)$ to measure the similarity between any two points and the average similarity, denoted by

$$S_k(\mathbf{X}_{\mathcal{I}_m}) = \frac{1}{m^2} \sum_{i \in \mathcal{I}_m} \sum_{j \in \mathcal{I}_m} k(\mathbf{x}_i, \mathbf{x}_j), \tag{2}$$

to measure the similarity between the selected samples. Obviously, $S(\mathbf{X}_{\mathcal{I}_m})$ can evaluate the diversity of $\mathbf{X}_{\mathcal{I}_m}$ since larger similarity implies smaller diversity.

As a statistical discrepancy which measures the distance between distributions, the maximum mean discrepancy (MMD) is introduced here to quantify the representativeness of $\mathbf{X}_{\mathcal{I}_m}$ to $\mathbf{X}_n$. Proposed by Gretton et al. [14], MMD is formally defined below:

**Definition 4.1 (Maximum Mean Discrepancy).** Let $P, Q$ be two Borel probability measures on $\mathcal{X}$. Suppose $f$ is sampled from the unit ball in a reproducing kernel Hilbert space (RKHS) $\mathcal{H}$ associated with its reproducing kernel $k(\cdot, \cdot)$, *i.e.,* $\|f\|_{\mathcal{H}} \leq 1$, then the MMD between $P$ and $Q$ is defined by

$$\mathrm{MMD}_k^2(P, Q) := \sup_{\|f\|_{\mathcal{H}} \leq 1} \left( \int f dP - \int f dQ \right)^2 = \mathbb{E}\left[ k(X, X') + k(Y, Y') - 2k(X, Y) \right], \tag{3}$$

where $X, X' \sim P$ and $Y, Y' \sim Q$ are independent copies.

We can next derive the empirical version for MMD that is able to measure the representativeness of $\mathbf{X}_{\mathcal{I}_m} = \{\mathbf{x}_i\}_{i \in \mathcal{I}_m}$ relative to $\mathbf{X}_n = \{\mathbf{x}_i\}_{i=1}^n$ by replacing $P, Q$ with the empirical distribution constructed by $\mathbf{X}_{\mathcal{I}_m}, \mathbf{X}_n$ in (3):

$$\mathrm{MMD}_k^2(\mathbf{X}_{\mathcal{I}_m}, \mathbf{X}_n) := \frac{1}{n^2} \sum_{i=1}^n \sum_{j=1}^n k(\mathbf{x}_i, \mathbf{x}_j) + \frac{1}{m^2} \sum_{i \in \mathcal{I}_m} \sum_{j \in \mathcal{I}_m} k(\mathbf{x}_i, \mathbf{x}_j) - \frac{2}{mn} \sum_{i=1}^n \sum_{j \in \mathcal{I}_m} k(\mathbf{x}_i, \mathbf{x}_j).$$
(4)

**Optimization objective.** Set $\mathrm{Rep}(\cdot, \cdot) = -\mathrm{MMD}_k^2(\cdot, \cdot)$ and $\mathrm{Div}(\cdot) = -S_k(\cdot)$ in (1), where $k$ is a proper kernel function, our optimization objective becomes

$$\min_{\mathcal{I}_m \subset [n]} \mathrm{MMD}_k^2(\mathbf{X}_{\mathcal{I}_m}, \mathbf{X}_n) + \lambda S_k(\mathbf{X}_{\mathcal{I}_m}). \tag{5}$$

Set $\lambda = \frac{1-\alpha}{\alpha m}$, since $\sum_{i=1}^n \sum_{j=1}^n k(\mathbf{x}_i, \mathbf{x}_j)$ is a constant, the objective function in (5) can be rewritten by

$$\alpha \, \mathrm{MMD}_k^2(\mathbf{X}_{\mathcal{I}_m}, \mathbf{X}_n) + \frac{1-\alpha}{m} S_k(\mathbf{X}_{\mathcal{I}_m}) + \frac{\alpha(\alpha-1)}{n^2} \sum_{i=1}^n \sum_{j=1}^n k(\mathbf{x}_i, \mathbf{x}_j)$$

$$= \frac{\alpha^2}{n^2} \sum_{i=1}^n \sum_{j=1}^n k(\mathbf{x}_i, \mathbf{x}_j) + \frac{1}{m^2} \sum_{i \in \mathcal{I}_m} \sum_{j \in \mathcal{I}_m} k(\mathbf{x}_i, \mathbf{x}_j) - \frac{2\alpha}{mn} \sum_{i=1}^n \sum_{j \in \mathcal{I}_m} k(\mathbf{x}_i, \mathbf{x}_j) \tag{6}$$

$$= \sup_{\|f\|_{\mathcal{H}} \leq 1} \left( \frac{1}{m} \sum_{i \in \mathcal{I}_m} f(\mathbf{x}_i) - \frac{\alpha}{n} \sum_{j=1}^n f(\mathbf{x}_j) \right)^2,$$

which defines a new concept called $\alpha$-MMD, denoted by $\mathrm{MMD}_{k,\alpha}(\mathbf{X}_{\mathcal{I}_m}, \mathbf{X}_n)$. This new concept distinguishes our method from those existing methods, which is essential for developing the sampling algorithms and theoretical analysis. Note that $\alpha$-MMD degenerates to classical MMD when $\alpha = 1$ and degenerates to average similarity when $\alpha = 0$. As $\alpha$ decreases, $\lambda$ increases, thereby encouraging the diversity for sample selection.

**Remark 1.** In the following context, all the kernels are assumed to be characteristic and positive definite if not specified. The following illustrates the advantages of the two properties.

**Characteristics kernels.** The MMD is generally a pseudo-metric on the space of all Borel probability distributions, implying that the MMD between two different distributions can be zero. Nevertheless, MMD becomes a proper metric when $k$ is a characteristic kernel, *i.e.,* $P \to \int_{\mathcal{X}} k(\cdot, \mathbf{x}) dP$ for any Borel probability distribution $P$ on $\mathcal{X}$ [29]. Therefore, MMD induced by characteristic kernels can be more appropriate for measuring representativeness.

**Positive definite kernels.** Aronszajn [2] showed that for every positive definite kernel $k(\cdot, \cdot)$, *i.e.,* its Gram matrix is always positive definite and symmetric, it uniquely determines an RKHS $\mathcal{H}$ and vice versa. This property is not only important for evaluating the property of MMD [43] but also required in optimizing MMD [32] by Frank-Wolfe algorithm.

## 4.2 Sampling Algorithm

In the previous research [36, 27, 50, 38, 58], sample selection is usually modelled by a non-convex combinatorial optimization problem. In contrast, following the idea of [4], we regard $\min_{\mathcal{I}_m \in [n]} \mathrm{MMD}_{k,\alpha}^2(\mathbf{X}_{\mathcal{I}_m}, \mathbf{X}_n)$ as a convex optimization problem by exploiting the convexity of $\alpha$-MMD, and then solve it by a fast iterative minimization procedure derived from Frank-Wolfe algorithm (see Appendix A for derivation details):

$$\mathbf{x}_{i_{p+1}^*} \in \arg\min_{i \in [n]} f_{\mathcal{I}_p^*}(\mathbf{x}_i), \mathcal{I}_{p+1}^* \leftarrow \mathcal{I}_p^* \cup \{i_{p+1}^*\}, \mathcal{I}_0 = \emptyset, \tag{7}$$

where $f_{\mathcal{I}_p}(\mathbf{x}_i) = \sum_{j \in \mathcal{I}_p} k(\mathbf{x}_i, \mathbf{x}_j) - \alpha p \sum_{l=1}^n k(\mathbf{x}_i, \mathbf{x}_l)$. As an extension of kernel herding [7], its corresponding algorithm (see Algorithm 2) is called Generalized Kernel Herding (GKH). Note that $f_{\mathcal{I}_p}(\mathbf{x}_i)$ is iteratively updated in Algorithm 2, which can save a lot of running time. However, GKH can select repeated samples that contradict the setting of sampling without replacement. To address this issue, we propose a modified iterating formula based on (7):

$$\mathbf{x}_{i_{p+1}^*} \in \arg\min_{i \in [n] \backslash \mathcal{I}_p^*} f_{\mathcal{I}_p^*}(\mathbf{x}_i), \mathcal{I}_{p+1}^* \leftarrow \mathcal{I}_p^* \cup \{i_{p+1}^*\}, \mathcal{I}_0^* = \emptyset, \tag{8}$$

**Algorithm 1** Generalized Kernel Herding without Replacement

---

**Require:** Data set $\mathbf{X}_n = \{\mathbf{x}_1, \cdots, \mathbf{x}_n\} \subset \mathcal{X}$; the number of selected samples $m < n$; a positive definite, characteristic and radial kernel $k(\cdot, \cdot)$ on $\mathcal{X} \times \mathcal{X}$; trade-off parameter $\alpha \leq 1$.
**Ensure:** Selected samples $\mathbf{X}_{\mathcal{I}_m^*} = \{\mathbf{x}_{i_1^*}, \cdots, \mathbf{x}_{i_m^*}\}$.
1: For each $\mathbf{x}_i \in \mathbf{X}_n$ calculate $\mu(\mathbf{x}_i) := \sum_{j=1}^{n} k(\mathbf{x}_j, \mathbf{x}_i)/n$.
2: Set $\beta_1 = 1$, $S_0 = 0$, $\mathcal{I} = \emptyset$.
3: **for** $p \in \{1, \cdots, m\}$ **do**
4:    $i_p^* \in \arg\min_{i \in [n] \backslash \mathcal{I}_p^*} S_{p-1}(\mathbf{x}_i) - \alpha\mu(\mathbf{x}_i)$
5:    For all $i \in [n] \backslash \mathcal{I}_p^*$, update $S_p(\mathbf{x}_i) = (1 - \beta_p)S_{p-1}(\mathbf{x}_i) + \beta_p k(\mathbf{x}_{i_p^*}, \mathbf{x}_i)$
6:    $\mathcal{I}_{p+1}^* \leftarrow \mathcal{I}_p^* \cup \{i_p^*\}$, $p \leftarrow p + 1$, set $\beta_p = 1/p$.
7: **end for**

---

which admits no repetitiveness in the selected samples. Its corresponding algorithm (see Algorithm 1) is thereby named as Generalized Kernel Herding without Replacement (GKHR), employed as the sampling algorithm for RDSS.

**Computational complexity.** Despite the time cost for calculating kernel functions, the computational complexity of GKHR is $O(mn)$, since in each iteration, the steps in lines 4 and 5 of Algorithm 2 respectively require $O(n)$ computations. Note that GKH has the same order of computational complexity as GKHR.

## 5 Theoretical Analysis

### 5.1 Generalization Bounds

Recall the core-set approach in [36], *i.e.,* for any $h \in \mathcal{H}$,

$$R(h) \leq \widehat{R}_S(h) + |R(h) - \widehat{R}_T(h)| + |\widehat{R}_T(h) - \widehat{R}_S(h)|,$$

where $T$ is the full labeled dataset and $S \subset T$ is the core set, $R(h)$ is the expected risk of $h$, $\widehat{R}_T(h), \widehat{R}_S(h)$ are empirical risk of $h$ on $T, S$. The first term $\widehat{R}_S(h)$ is unknown before we label the selected samples, and the second term $|R(h) - \widehat{R}_T(h)|$ can be upper bounded by the so-called generalization bounds [3, 64] which do not depend on the choice of core set. Therefore, to control the upper bound of $R(h)$, we only need to analyse the upper bound of the third term $|\widehat{R}_T(h) - \widehat{R}_S(h)|$ called core-set loss, which requires several mild assumptions. Shalit, et al. [37] derived a MMD-type upper bound for $|\widehat{R}_T(h) - \widehat{R}_S(h)|$ to estimate individual treatment effect, while our bound is generalized to a wider range of tasks.

Let $\mathcal{H}_1 = \{h | h : \mathcal{X} \to \mathcal{Y}\}$ be a hypothesis set in which we are going to select a predictor and suppose that the labelled data $T = \{(\mathbf{x}_i, y_i)\}_{i=1}^n$ are i.i.d. sampled from a random vector $(X, Y)$ defined on $\mathcal{X} \times \mathcal{Y}$. We firstly assume that $\mathcal{H}_1$ is an RKHS, which is mild in machine learning theory [3, 5].

**Assumption 5.1.** $\mathcal{H}_1$ is an RKHS associated with bounded positive definite kernel $k_1$ where the norm of any $h \in \mathcal{H}_1$ is bounded by $K_h$.

We further make RKHS assumptions on the functional space of $\mathbb{E}(Y|X)$ and $\text{Var}(Y|X)$ that are fundamental in the field of conditional distribution embedding [41, 43].

**Assumption 5.2.** There is an RKHS $\mathcal{H}_2$ associated with bounded positive definite kernel $k_2$ such that $\mathbb{E}(Y|X) \in \mathcal{H}_2$ and the norm of any $\mathbb{E}(Y|X)$ is bounded by $K_m$.

**Assumption 5.3.** There is an RKHS $\mathcal{H}_3$ associated with bounded positive definite kernel $k_3$ such that $\text{Var}(Y|X) \in \mathcal{H}_3$ and the norm of any $\text{Var}(Y|X)$ is bounded by $K_s$.

We next give a $\alpha$-MMD-type upper bound for the core-set loss by the following theorem:

**Theorem 5.4.** *Take $k = k_1^2 + k_1 k_2 + k_3$, then under assumptions 1-3, for any selected samples $S \subset T$, there exists a positive constant $K_c$ such that the following inequality holds:*

$$|\widehat{R}_T(h) - \widehat{R}_S(h)| \leq K_c(\text{MMD}_{k,\alpha}(\mathbf{X}_S, \mathbf{X}_T) + (1 - \alpha)\sqrt{K})^2,$$

*where $0 \leq \alpha \leq 1$, $0 \leq \max_{\mathbf{x} \in \mathcal{X}} k(\mathbf{x}, \mathbf{x}) = K$ and $\mathbf{X}_S, \mathbf{X}_T$ are projections of $S, T$ on $\mathcal{X}$.*

Therefore, minimizing $\alpha$-MMD can optimize the generalization bound for $R(h)$ and benefit the generalizability of the trained model (predictor).

## 5.2 Finite-Sample-Error-Bound for GKHR

The concept of convergence does not apply to analyzing GKHR. With $n$ fixed, GKHR iterates for at most $n$ times and then returns $\mathbf{X}_{\mathcal{I}_n^*} = \mathbf{X}_n$. Consequently, we analyze the performance of GKHR by its finite-sample-error bound. Previous to that, we make an assumption on the mean of $f_{\mathcal{I}_p^*}$ over the full unlabeled dataset.

**Assumption 5.5.** For any $\mathcal{I}_p^*$ returned by GKHR, $1 \leq p \leq m-1$, there exists $p+1$ elements $\{\mathbf{x}_{j_l}\}_{l=1}^{p+1}$ in $\mathbf{X}_n$ such that

$$f_{\mathcal{I}_p^*}(\mathbf{x}_{j_1}) \leq \cdots f_{\mathcal{I}_p^*}(\mathbf{x}_{j_{p+1}}) \leq \frac{\sum_{i=1}^n f_{\mathcal{I}_p^*}(\mathbf{x}_i)}{n}.$$

When $m$ is not relatively small, this assumption is rather unrealistic. Nevertheless, under our low-budget setting, especially when $m \ll n$, the assumption becomes an extension of the principle that "the minimum is never larger than the mean", which still probably makes sense. We can then show that the decaying rate for optimization error of GKHR can be upper bounded by $O(\log m/m)$:

**Theorem 5.6.** *Let $\mathbf{X}_{\mathcal{I}_m^*}$ be the samples selected by GKHR, under assumption 4, it holds that*

$$\mathrm{MMD}_{k,\alpha}^2\left(\mathbf{X}_{\mathcal{I}_m^*}, \mathbf{X}_n\right) \leq C_\alpha^2 + B\frac{2 + \log m}{m+1} \tag{9}$$

*where $B = 2K$, $0 \leq \max_{\mathbf{x} \in \mathcal{X}} k(\mathbf{x}, \mathbf{x}) = K$, $C_\alpha^2 = (1-\alpha)^2\overline{K}$ where $\overline{K}$ is defined in Lemma B.6.*

## 6 Choice of Kernel and Hyperparameter Tuning

In this section, we make some suggestions for choosing the kernel and tuning the hyperparameter $\alpha$.

**Choice of kernel.** Recall Remark 1 in Section 4.1, we only consider characteristic and positive definite kernels in RDSS. Since the Gaussian kernels are the most commonly used kernels in the field of machine learning and statistics [3, 15], we introduce Gaussian kernel as our choice, which is defined by $k(\mathbf{x}, \mathbf{y}) = \exp(-\|\mathbf{x} - \mathbf{y}\|_2^2/\sigma^2)$. The bandwidth parameter $\sigma$ is set to be the median distance between samples in the aggregate dataset [15], *i.e.*, $\sigma = \mathrm{Median}(\{\|\mathbf{x} - \mathbf{y}\|_2 | \mathbf{x}, \mathbf{y} \in \mathbf{X}_n\})$, since the median is robust and also compromises between extreme cases.

**Tuning trade-off hyperparameter $\alpha$.** According to Theorem 5.6 and Lemma B.3, by straightforward deduction we have

$$\mathrm{MMD}_k\left(\mathbf{X}_{\mathcal{I}_m^*}, \mathbf{X}_n\right) \leq C_\alpha + \mathcal{O}\left(\sqrt{\frac{\log m}{m}}\right) + (1-\alpha)\sqrt{K}$$

to upper bound the MMD between the selected samples and the full dataset under a low-budget setting. We can just set $\alpha \in [1 - \frac{1}{\sqrt{m}}, 1)$ so that the upper bound of the MMD would not be larger than the one of $\alpha$-MMD in the perspective of the order of magnitude.

## 7 Experiments

In this section, we first explain the implementation details of our method RDSS in Section 7.1. Next, we compare RDSS with other sampling methods by integrating them into two state-of-the-art (SOTA) SSL approaches (FlexMatch [63] and Freematch [53]) on five datasets (CIFAR-10/100, SVHN, STL-10 and ImageNet-1k) in Section 7.2. The details of the datasets, the visualization results and the computational complexity of different sampling methods are shown in Appendix D.2, D.3, and D.4, respectively. We also compare against various AL/SSAL approaches in Section 7.3. Lastly, we make quantitative analyses of the trade-off parameter $\alpha$ in Section 7.4.

## 7.1 Implementation Details of Our Method

First, we leverage the pre-trained image feature extraction capabilities of CLIP [33], a vision transformer architecture, to extract features. Subsequently, the [CLS] token features produced by the model's final output are employed for sample selection. During the sample selection phase, the Gaussian kernel function is chosen as the kernel method to compute the similarity of samples in an infinite-dimensional feature space. The value of $\sigma$ for the Gaussian kernel function is set as explained in Section 6. To ensure diversity in the sampled data, we introduce a penalty factor given by $\alpha = 1 - \frac{1}{\sqrt{m}}$, where $m$ denotes the number of selected samples. Concretely, we set $m = \{40, 250, 4000\}$ for CIFAR-10, $m = \{400, 2500, 10000\}$ for CIFAR-100, $m = \{250, 1000\}$ for SVHN, $m = \{40, 250\}$ for STL-10 and $m = \{100000\}$ for ImageNet. Next, the selected samples are used for two SSL approaches, which are trained and evaluated on the datasets using the codebase Unified SSL Benchmark (USB) [52]. The optimizer for all experiments is standard stochastic gradient descent (SGD) with a momentum of 0.9 [44]. The initial learning rate is 0.03 with a learning rate decay of 0.0005. We use ResNet-50 [16] for the ImageNet experiment and Wide ResNet-28-2 [62] for other datasets. Finally, we evaluate the performance with the Top-1 classification accuracy metric on the test set. Experiments are run on 8*NVIDIA Tesla A100 (40 GB) and 2*Intel 6248R 24-Core Processor. We average our results over five independent runs.

## 7.2 Comparison with Other Sampling Methods

**Main results** We apply RDSS on Flexmatch and Freematch to compare with the following three baselines and two SOTA methods in SSL under different annotation budget settings. The baselines conclude **Stratified**, **Random** and $k$-**Means**, while the two SOTA methods are **USL** [50] and **ActiveFT** [57]. The results are shown on Table 1 from which we have several observations: (1) Our proposed RDSS achieves the highest accuracy, outperforming other sampling methods, which underscores the effectiveness of our approach; (2) USL attains suboptimal results under most budget settings yet exhibits a significant gap compared to RDSS, particularly under severely constrained ones. For instance, FreeMatch achieves a 4.95% rise on the STL-10 with a budget of 40; (3) In most experiments, RDSS either approaches or surpasses the performance of stratified sampling, especially on SVHN and STL-10. However, the stratified sampling method is practically infeasible given that the category labels of the data are not known a priori.

**Results on ImageNet** We also compare the second-best method USL with RDSS on ImageNet. Following the settings of FreeMatch [53], we select 100k samples for annotation. FreeMatch, using RDSS and USL as sampling methods, achieves 58.24% and 56.86% accuracy, respectively, demonstrating a substantial enhancement in the performance of our method over the USL approach.

Table 1: Comparison with other sampling methods. Due to stratified sampling limitations, the results are marked in grey. Top and second-best performances are bolded and underlined, respectively, excluding stratified sampling. Metrics represent mean accuracy and standard deviation over five independent runs.

| Dataset | CIFAR-10 | | | CIFAR-100 | | | SVHN | | STL-10 | |
|---|---|---|---|---|---|---|---|---|---|---|
| Budget | 40 | 250 | 4000 | 400 | 2500 | 10000 | 250 | 1000 | 40 | 250 |
| *Applied to FlexMatch [63]* | | | | | | | | | | |
| Stratified | 91.45±3.41 | 95.10±0.25 | 95.63±0.24 | 50.23±0.41 | 67.38±0.45 | 73.61±0.43 | 89.60±1.86 | 93.66±0.49 | 75.33±3.74 | 92.29±0.64 |
| Random | 87.30±4.61 | 93.95±0.91 | 95.17±0.59 | 45.58±0.97 | 66.48±0.98 | 72.61±0.83 | 87.67±1.16 | 94.06±1.14 | 65.81±1.21 | 90.70±0.79 |
| $k$-Means | 81.23±8.71 | 94.59±0.51 | 95.09±0.65 | 41.60±1.24 | 65.99±0.57 | 71.53±0.42 | 90.28±0.69 | 93.82±1.04 | 55.43±0.39 | 90.64±1.05 |
| USL [50] | 91.73±0.13 | 94.89±0.20 | 95.43±0.15 | 46.89±0.46 | 66.75±0.37 | 72.53±0.32 | 90.03±0.63 | 93.10±0.78 | 75.65±0.60 | 90.77±0.36 |
| ActiveFT [57] | 70.87±4.14 | 93.85±1.37 | 95.31±0.75 | 25.69±0.64 | 57.19±2.06 | 70.96±0.75 | 89.32±1.87 | 92.53±0.43 | 55.57±1.42 | 87.28±1.19 |
| RDSS (Ours) | **94.69±0.28** | **95.21±0.47** | **95.71±0.10** | **48.12±0.36** | 67.27±0.55 | **73.21±0.29** | **91.70±0.39** | **95.70±0.35** | **77.96±0.52** | **93.16±0.41** |
| *Applied to FreeMatch [53]* | | | | | | | | | | |
| Stratified | 95.05±0.15 | 95.40±0.23 | 95.80±0.29 | 51.29±0.56 | 67.69±0.58 | 73.90±0.53 | 92.58±1.05 | 94.22±0.78 | 79.16±5.01 | 91.36±0.18 |
| Random | 93.41±1.24 | 93.98±0.91 | 95.56±0.17 | 47.16±1.25 | 66.09±1.08 | 72.09±0.99 | 91.62±1.88 | 94.40±1.28 | 76.66±2.43 | 90.72±0.97 |
| $k$-Means | 88.05±5.07 | 94.80±0.48 | 95.51±0.37 | 44.07±1.94 | 66.09±0.39 | 71.69±0.72 | 93.30±0.46 | 94.68±0.72 | 63.22±4.92 | 89.99±0.87 |
| USL [50] | 93.81±0.62 | 95.19±0.18 | 95.78±0.29 | 47.07±0.78 | 66.92±0.33 | 72.59±0.36 | 93.36±0.53 | 94.44±0.44 | 76.95±0.86 | 90.58±0.58 |
| ActiveFT [57] | 78.13±2.87 | 94.54±0.81 | 95.33±0.53 | 26.67±0.46 | 56.23±0.85 | 71.20±0.68 | 92.60±0.51 | 93.71±0.54 | 63.31±2.99 | 86.60±0.30 |
| RDSS (Ours) | **95.05±0.13** | **95.50±0.20** | **95.98±0.28** | **48.41±0.59** | **67.40±0.23** | **73.13±0.19** | **94.54±0.46** | **95.83±0.37** | **81.90±1.72** | **92.22±0.40** |

## 7.3 Comparison with AL/SSAL Approaches

First, we compare RDSS against various traditional AL approaches on CIFAR-10/100. AL approaches conclude **CoreSet** [36], **VAAL** [39], **LearnLoss** [60] and **MCDAL** [8]. For a fair comparison, we

exclusively use samples selected by RDSS for supervised learning compared to other AL approaches, considering that AL relies solely on labelled samples for supervised learning. The implementation details are shown in Appendix D.5. The experimental results are presented in Table 2, from which we observe that RDSS achieves the highest accuracy under almost all budget settings when relying solely on labelled data for supervised learning, with notable improvements on CIFAR-100.

Second, we compare RDSS with sampling methods used in SSAL when applied to the same SSL framework (*i.e.,* FlexMatch or FreeMatch) on CIFAR-10. The sampling methods conclude **Core-SetSSL** [36], **MMA** [42], **CBSSAL** [12], and **TOD-Semi** [17]. In detail, we tune recent SSAL approaches with their public implementations and run experiments under an extremely low-budget setting, *i.e.,* 40 samples in a 20-random-and-20-selected setting. Table 3 illustrates that the performance of most SSAL approaches falls below that of random sampling methods under extremely low-budget settings. This inefficiency stems from the dependency of sample selection on model performance within the SSAL framework, which struggles when the model is weak. Our model-free method, in contrast, selects samples before training, avoiding these pitfalls.

Table 2: Comparison with AL approaches under Supervised Learning (SL) paradigm. The best performance is bold and the second best performance is underlined.

| Dataset | CIFAR-10 | | CIFAR-100 | |
|---|---|---|---|---|
| Budget | 7500 | 10000 | 7500 | 10000 |
| CoreSet | 85.46 | 87.56 | 47.17 | 53.06 |
| VAAL | 86.82 | 88.97 | 47.02 | 53.99 |
| LearnLoss | 85.49 | 87.06 | 47.81 | 54.02 |
| MCDAL | **87.24** | 89.40 | 49.34 | 54.14 |
| SL+RDSS (Ours) | 87.18 | **89.77** | **50.13** | **56.04** |
| Whole Dataset | 95.62 | | 78.83 | |

Table 3: Comparison with SSAL approaches. The green (red) arrow represents the improvement (decrease) compared to the random sampling method.

| Method | FlexMatch | FreeMatch |
|---|---|---|
| Stratified | 91.45 | 95.05 |
| Random | 87.30 | 93.41 |
| CoreSetSSL | 87.66 ↑ 0.36 | 91.24 ↓ 2.17 |
| MMA | 74.61 ↓ 12.69 | 87.37 ↓ 6.04 |
| CBSSAL | 86.58 ↓ 0.72 | 91.68 ↓ 1.73 |
| TOD-Semi | 86.21 ↓ 1.09 | 90.77 ↓ 2.64 |
| RDSS (Ours) | **94.69** ↑ 7.39 | **95.05** ↑ 1.64 |

Third, we directly compare RDSS with the above AL/SSAL approaches when applied to SSL, which may better reflect the paradigm differences. The experimental results and analysis are in the Appendix D.6.

### 7.4 Trade-off Parameter $\alpha$

We analyze the effect of different $\alpha$ with Freematch on CIFAR-10/100. The results are presented in Table 4, from which we have several observations: (1) Our proposed RDSS achieves the highest accuracy under all budget conditions, surpassing those that employ a fixed value; (2) The $\alpha$ that achieve the best or the second best performance are within the interval we set, which is in line with our theoretical derivation in Section 6; (3) The experimental outcomes exhibit varying degrees of reduction compared to our approach when the representativeness or diversity term is removed.

Table 4: Effect of different $\alpha$. The grey results indicate that the $\alpha$ is outside the interval we set in Section 6, *i.e.,* $\alpha < 1 - 1/\sqrt{m}$, while the black results indicate that the $\alpha$ is within the interval we set, *i.e.,* $1 - 1/\sqrt{m} \leq \alpha \leq 1$. Among them, $\alpha = 0$ and $\alpha = 1$ indicate the removal of the representativeness and diversity terms, respectively. The best performance is bold, and the second-best performance is underlined.

| Dataset | CIFAR-10 | | | CIFAR-100 | | |
|---|---|---|---|---|---|---|
| Budget ($m$) | 40 | 250 | 4000 | 400 | 2500 | 10000 |
| 0 | 85.54±0.48 | 93.55±0.34 | 94.58±0.27 | 39.26±0.52 | 63.77±0.26 | 71.90±0.17 |
| 0.40 | 92.28±0.24 | 93.68±0.13 | 94.95±0.12 | 42.56±0.47 | 65.88±0.24 | 71.71±0.29 |
| 0.80 | 94.42±0.49 | 94.94±0.37 | 95.15±0.35 | 45.62±0.35 | 66.87±0.20 | 72.45±0.23 |
| 0.90 | 94.33±0.28 | 95.03±0.21 | 95.20±0.42 | 48.12±0.50 | 67.14±0.16 | 72.15±0.23 |
| 0.95 | 94.44±0.64 | 95.07±0.26 | 95.45±0.38 | **48.41**±0.59 | 67.11±0.29 | 72.80±0.35 |
| 0.98 | 94.51±0.39 | 95.02±0.15 | 95.31±0.44 | 48.33±0.54 | **67.40**±0.23 | 72.68±0.22 |
| 1 | 94.53±0.42 | 95.01±0.23 | 95.54±0.25 | 48.18±0.36 | 67.20±0.29 | 73.05±0.18 |
| $1 - 1/\sqrt{m}$ (Ours) | **95.05**±0.13 | **95.50**±0.20 | **95.98**±0.28 | **48.41**±0.59 | **67.40**±0.23 | **73.13**±0.19 |

# 8   Conclusion

In this work, we propose a model-free sampling method, RDSS, to select a subset from unlabeled data for annotation in SSL. The primary innovation of our approach lies in the introduction of $\alpha$-MMD, designed to evaluate the representativeness and diversity of selected samples. Under a low-budget setting, we develop a fast and efficient algorithm GKHR for this problem using the Frank-Wolfe algorithm. Both theoretical analyses and empirical experiments demonstrate the effectiveness of RDSS. In future research, we would like to apply our methodology to scenarios where labelling is cost-prohibitive, such as in the medical domain.

## Acknowledgements

This research was partially supported by National Natural Science Foundation of China under grant No. 82202984, Zhejiang Key R&D Program of China under grants No. 2023C03053 and No. 2024SSYS0026, and US National Science Foundation under grant No. 2316011. We thank Prof. Fred Hickernell and Mr. Yulong Wan for offering useful comments on this paper.

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

# A   Algorithms

## A.1   Derivation of Generalized Kernel Herding (GKH)

*Proof.* The proof technique is borrowed from [32]. Let us firstly define a weighted modification of $\alpha$-MMD. For any $\mathbf{w} \in \mathbb{R}^n$ such that $\mathbf{w}^\top \mathbf{1} = 1$, the weighted $\alpha$-MMD is defined by

$$\text{MMD}^2_{k,\alpha,\mathbf{X}_n}(\mathbf{w}) = \mathbf{w}^\top \mathbf{K} \mathbf{w} - 2\alpha \mathbf{w}^\top \mathbf{p} + \alpha^2 \overline{K},$$

where $\mathbf{K} = [k(\mathbf{x}_i, \mathbf{x}_j)]_{1 \leq i,j \leq n}$, $\overline{K} = \mathbf{1}^\top \mathbf{K} \mathbf{1}/n^2$, $\mathbf{p} = (\mathbf{e}_1^\top \mathbf{K} \mathbf{1}/n, \cdots, \mathbf{e}_n^\top \mathbf{K} \mathbf{1}/n)$, $\{\mathbf{e}_i\}_{i=1}^n$ is the set of standard basis of $\mathbb{R}^n$. It is obvious that for any $\mathcal{I}_p \subset [n]$,

$$\text{MMD}^2_{k,\alpha,\mathbf{X}_n}(\mathbf{w}_p) = \text{MMD}^2_{k,\alpha}(\mathbf{X}_{\mathcal{I}_p}, \mathbf{X}_n),$$

where $(\mathbf{w}_p)_i = 1/p$ if $i \in \mathcal{I}_p$, and $(\mathbf{w}_p)_i = 0$ if not. Therefore, weighted $\alpha$-MMD is indeed a generalization of $\alpha$-MMD. Let

$$\mathbf{K}_* = \mathbf{K} - 2\alpha \mathbf{p} \mathbf{1}^\top + \alpha^2 \overline{K} \mathbf{1} \mathbf{1}^\top$$

we obtain the quadratic form expression of weighted $\alpha$-MMD by $\text{MMD}^2_{k,\alpha,\mathbf{X}_n}(\mathbf{w}) = \mathbf{w}^\top \mathbf{K}_* \mathbf{w}$, where $\mathbf{K}_*$ is strictly positive definite if the unlabeled data are pairwise different, $\mathbf{w} \neq \mathbf{w}_n$ and $k$ is a characteristic kernel according to [32].

Recall our low-budget setting (so $\mathbf{w} \neq \mathbf{w}_n$ holds) and assumption for kernel, $\mathbf{K}_*$ is indeed a strictly positive definite matrix. Thus $\text{MMD}^2_{k,\alpha,\mathbf{X}_n}$ is a convex functional w.r.t. $\mathbf{w}$, leading to the fact that $\min_{\mathbf{w}^\top \mathbf{1} = 1} \text{MMD}^2_{k,\alpha,\mathbf{X}_n}(\mathbf{w})$ can be solved by Frank-Wolfe algorithm. Then for $1 \leq p < n$,

$$\mathbf{s}_p \in \underset{\mathbf{s}^\top \mathbf{1} = 1}{\arg\min} \, \mathbf{s}^\top (\mathbf{K} \mathbf{w}_p - \alpha \mathbf{p}) = \underset{\mathbf{e}_i, i \in [n]}{\arg\min} \, \mathbf{e}_i^\top (\mathbf{K} \mathbf{w}_p - \alpha \mathbf{p}).$$

Let $\mathbf{e}_{i_p^*} = \mathbf{s}_p$, under uniform step size in Frank-Wolfe algorithm, we have

$$\mathbf{w}_{p+1} = \left( \frac{p}{p+1} \right) \mathbf{w}_p + \frac{1}{p+1} \mathbf{e}_{i_p^*}, \mathbf{w}_0 = 0$$

as the update formula of Frank-Wolfe algorithm, which is equivalent to

$$i_p^* \in \arg\min_{i \in [n]} \sum_{j \in \mathcal{I}_p} k(\mathbf{x}_i, \mathbf{x}_j) - \alpha p \sum_{l=1}^n k(\mathbf{x}_i, \mathbf{x}_l).$$

then we can immediately derive the iterating formula in (7).   □

## A.2   Pseudo Codes

---
**Algorithm 2** Generalized Kernel Herding
---
**Require:** Data set $\mathbf{X}_n = \{\mathbf{x}_1, \cdots, \mathbf{x}_n\} \subset \mathcal{X}$; the number of selected samples $m < n$; a positive definite, characteristic and radial kernel $k(\cdot, \cdot)$ on $\mathcal{X} \times \mathcal{X}$; trade-off parameter $\alpha \leq 1$.
**Ensure:** selected samples $\mathbf{X}_{\mathcal{I}_m^*} = \{\mathbf{x}_{i_1^*}, \cdots, \mathbf{x}_{i_m^*}\}$.
 1: For each $\mathbf{x}_i \in \mathbf{X}_n$ calculate $\mu(\mathbf{x}_i) := \sum_{j=1}^n k(\mathbf{x}_j, \mathbf{x}_i)/n$.
 2: Set $\beta_1 = 1$, $S_0 = 0$, $\mathcal{I} = \emptyset$.
 3: **for** $p \in \{1, \cdots, m\}$ **do**
 4:    $i_p^* \in \arg\min_{i \in [n]} S_{p-1}(\mathbf{x}_i) - \alpha \mu(\mathbf{x}_i)$
 5:    For all $i \in [n]$, update $S_p(\mathbf{x}_i) = (1 - \beta_p) S_{p-1}(\mathbf{x}_i) + \beta_p k(\mathbf{x}_{i_p^*}, \mathbf{x}_i)$
 6:    $\mathcal{I}_{p+1}^* \leftarrow \mathcal{I}_p^* \cup \{i_p^*\}$, $p \leftarrow p + 1$, set $\beta_p = 1/p$.
 7: **end for**
---

# B   Technical Lemmas

**Lemma B.1** (Lemma 2 [32])**.** *Let $(t_k)_k$ and $(\alpha_k)_k$ be two real positive sequences and $A$ be a strictly positive real. If $t_k$ satisfies*

$$t_1 \leq A \text{ and } t_{k+1} \leq (1 - \alpha_{k+1}) t_k + A \alpha_{k+1}^2, k \geq 1,$$

*with $\alpha_k = 1/k$ for all $k$, then $t_k < A(2 + \log k)/(k+1)$ for all $k > 1$.*

**Lemma B.2.** *The selected samples $\mathbf{X}_{\mathcal{I}_m^*}$ generated by GKH (Algorithm 2) satisfies*

$$\text{MMD}_{k,\alpha}^2 \left( \mathbf{X}_{\mathcal{I}_m^*}, \mathbf{X}_n \right) \le M_\alpha^2 + B \frac{2 + \log m}{m + 1} \tag{10}$$

*where $B = 2K$, $0 \le \max_{\mathbf{x} \in \mathcal{X}} k(\mathbf{x}, \mathbf{x}) \le K$, $M_\alpha^2$ is defined by*

$$M_\alpha^2 := \min_{\mathbf{w}^\top \mathbf{1} = 1, \mathbf{w} \ge 0} \text{MMD}_{k,\alpha,\mathbf{X}_n}^2 (\mathbf{w})$$

*Proof.* Following the notations in Appendix A, let $\mathbf{p}_\alpha = \alpha \mathbf{p}$, we could straightly follow the proof for finite-sample-size error bound of kernel herding with predefined step sizes given by [32] to derive Lemma B.2, without any other technique. The detailed proof is omitted. $\qquad\square$

**Lemma B.3.** *Let $\mathcal{H}$ be an RKHS over $\mathcal{X}$ associated with positive definite kernel $k$, and $0 \le \max_{\mathbf{x} \in \mathcal{X}} k(\mathbf{x}, \mathbf{x}) \le K$. Let $\mathbf{X}_m = \{\mathbf{x}_i\}_{i=1}^m$, $\mathbf{Y}_n = \{\mathbf{y}_j\}_{j=1}^m$, $\mathbf{x}_i, \mathbf{y}_j \in \mathcal{X}$. Then for any $\alpha \le 1$,*

$$| \text{MMD}_{k,\alpha}(\mathbf{X}_m, \mathbf{Y}_n) - \text{MMD}_k(\mathbf{X}_m, \mathbf{Y}_n)| \le (1 - \alpha)\sqrt{K}$$

*Proof.*

$$
\begin{aligned}
&|\text{MMD}_{k,\alpha}(\mathbf{X}_m, \mathbf{Y}_n) - \text{MMD}_k(\mathbf{X}_m, \mathbf{Y}_n)| \\
&= \left| \sup_{\|f\|_\mathcal{H} \le 1} \left( \frac{1}{m} \sum_{i=1}^m f(\mathbf{x}_i) - \frac{\alpha}{n} \sum_{j=1}^n f(\mathbf{y}_j) \right) - \sup_{\|f\|_\mathcal{H} \le 1} \left( \frac{1}{m} \sum_{i=1}^m f(\mathbf{x}_i) - \frac{1}{n} \sum_{j=1}^n f(\mathbf{y}_j) \right) \right| \\
&\le \sup_{\|f\|_\mathcal{H} \le 1} \left| \frac{1 - \alpha}{n} \sum_{i=1}^n f(y_i) \right| = \left( \frac{1 - \alpha}{n} \right) \sup_{\|f\|_\mathcal{H} \le 1} \left| \sum_{i=1}^n f(y_i) \right| \\
&= \left( \frac{1 - \alpha}{n} \right) \sup_{\|f\|_\mathcal{H} \le 1} \left| \sum_{j=1}^n \langle f, k(\cdot, \mathbf{y}_j) \rangle_\mathcal{H} \right| \le \left( \frac{1 - \alpha}{n} \right) \sup_{\|f\|_\mathcal{H} \le 1} \sum_{j=1}^n |\langle f, k(\cdot, \mathbf{y}_j) \rangle_\mathcal{H}| \\
&\le \left( \frac{1 - \alpha}{n} \right) \sup_{\|f\|_\mathcal{H} \le 1} \sum_{j=1}^n \|f\|_\mathcal{H} \|k(\cdot, \mathbf{y}_j)\|_\mathcal{H} \le (1 - \alpha)\sqrt{K}.
\end{aligned}
$$

$\qquad\square$

**Lemma B.4** (Proposition 12.31 [48]). *Suppose that $\mathcal{H}_1$ and $\mathcal{H}_2$ are reproducing kernel Hilbert spaces of real-valued functions with domains $\mathcal{X}_1$ and $\mathcal{X}_2$, and equipped with kernels $k_1$ and $k_2$, respectively. Then the tensor product space $\mathcal{H} = \mathcal{H}_1 \otimes \mathcal{H}_2$ is an RKHS of real-valued functions with domain $\mathcal{X}_1 \times \mathcal{X}_2$, and with kernel function*

$$k\left((x_1, x_2), (x_1', x_2')\right) = k_1(x_1, x_1') k_2(x_2, x_2').$$

**Lemma B.5** (Theorem 5.7 [31]). *Let $f \in \mathcal{H}_1$ and $g \in \mathcal{H}_2$, where $\mathcal{H}_1, \mathcal{H}_2$ be two RKHS containing real-valued functions on $\mathcal{X}$, which is associated with positive definite kernel $k_1, k_2$ and canonical feature map $\phi_1, \phi_2$, then for any $x \in \mathcal{X}$,*

$$f(x) + g(x) = \langle f, \phi_1(x) \rangle_{\mathcal{H}_1} + \langle g, \phi_2(x) \rangle_{\mathcal{H}_2} = \langle f + g, (\phi_1 + \phi_2)(x) \rangle_{\mathcal{H}_1 + \mathcal{H}_2},$$

*where*

$$\mathcal{H}_1 + \mathcal{H}_2 = \{f_1 + f_2 | f_i \in \mathcal{H}_i\}$$

*and $\phi_1 + \phi_2$ is the canonical feature map of $\mathcal{H}_1 + \mathcal{H}_2$. Furthermore,*

$$\|f + g\|_{\mathcal{H}_1 + \mathcal{H}_2}^2 \le \|f\|_{\mathcal{H}_1}^2 + \|g\|_{\mathcal{H}_2}^2.$$

**Lemma B.6.** *For any unlabeled dataset $\mathbf{X}_n \subset \mathcal{X}$ and any subset $\mathbf{X}_{\mathcal{I}_m}$,*

$$\text{MMD}_{k,\alpha}^2(\mathbf{X}_n, \mathbf{X}_n) = (1 - \alpha)^2 \overline{K}, \text{MMD}_{k,\alpha}^2(\mathbf{X}_{\mathcal{I}_m}, \mathbf{X}_n) \le (1 + \alpha^2)K,$$

*where $\overline{K} = \sum_{i=1}^n \sum_{j=1}^n k(\mathbf{x}_i, \mathbf{x}_j)/n^2$, $K = \max_{\mathbf{x} \in \mathcal{X}} k(\mathbf{x}, \mathbf{x})$.*

Lemma B.6 is directly derived from the definition of $\alpha$-MMD.

## C Proof of Theorems

*Proof for Theorem 5.4.* The proof borrows the technique introduced in [37] for decomposing the expected risk of hypotheses.

Firstly, let us denote that $\mathcal{H}_4 = \mathcal{H}_1 \otimes \mathcal{H}_1 + \mathcal{H}_1 \otimes \mathcal{H}_2 + \mathcal{H}_3$, with kernel $k_4 = k_1^2 + k_1 k_2 + k_3$ and canonical feature map $\phi_4 = \phi_1 \otimes \phi_1 + \phi_1 \otimes \phi_2 + \phi_3$.

Under the assumptions in Theorem 5.4, according to Theorem 4 in [41], we have for any $\mathbf{x} \in \mathcal{X}$,

$$h(\mathbf{x}) = \langle h, \phi_1(\mathbf{x}) \rangle_{\mathcal{H}_1}, \mathbb{E}[Y|\mathbf{x}] = \langle \mathbb{E}[Y|X], \phi_2(\mathbf{x}) \rangle_{\mathcal{H}_2},$$

$$\mathrm{Var}(Y|\mathbf{x}) = \langle \mathrm{Var}(Y|X), \phi_3(\mathbf{x}) \rangle_{\mathcal{H}_3}$$

where $\phi_1, \phi_2, \phi_3$ are canonical feature maps in $\mathcal{H}_1, \mathcal{H}_2, \mathcal{H}_3$. Denote that $m = \mathbb{E}[Y|X]$ and $s = \mathrm{Var}(Y|X)$. Now by definition,

$$R(h) = \mathbb{E}\left[\ell(h(\mathbf{x}), y)\right] = \int_{\mathcal{X}} \int_{\mathcal{Y}} \ell(h(\mathbf{x}), y) p(y|\mathbf{x}) p(\mathbf{x}) d\mathbf{x} dy = \int_{\mathcal{X}} f(\mathbf{x}) p(\mathbf{x}) d\mathbf{x}$$

where

$$\begin{aligned}
f(x) &= \int_{\mathcal{Y}} (y - h(\mathbf{x}))^2 p(y|\mathbf{x}) dy \\
&= \mathrm{Var}(Y|\mathbf{x}) - 2h(\mathbf{x})\mathbb{E}[Y|\mathbf{x}] + h^2(\mathbf{x}) \\
&= \langle s, \phi_3(\mathbf{x}) \rangle_{\mathcal{H}_3} - 2 \langle h, \phi_1(\mathbf{x}) \rangle_{\mathcal{H}_1} \langle m, \phi_2(\mathbf{x}) \rangle_{\mathcal{H}_2} + \langle h, \phi_1(\mathbf{x}) \rangle_{\mathcal{H}_1} \langle h, \phi_1(\mathbf{x}) \rangle_{\mathcal{H}_1} \\
&= \langle s, \phi_3(\mathbf{x}) \rangle_{\mathcal{H}_3} - \langle 2h \otimes m, (\phi_1 \otimes \phi_2)(\mathbf{x}) \rangle_{\mathcal{H}_1 \otimes \mathcal{H}_2} + \langle h \otimes h, (\phi_1 \otimes \phi_1)(\mathbf{x}) \rangle_{\mathcal{H}_1 \otimes \mathcal{H}_1} \\
&= \langle s - 2h \otimes m + h \otimes h, \phi_4(x) \rangle_{\mathcal{H}_4}
\end{aligned}$$

where the fourth equality holds by Lemma B.4 and the last equality holds by Lemma B.5, then $f \in \mathcal{H}_4$, and

$$\begin{aligned}
\|f\|_{\mathcal{H}_4} &= \|s - 2h \otimes m + h \otimes h\|_{\mathcal{H}_4} \\
&\leq \|s\|_{\mathcal{H}_4} + \|2h \otimes m\|_{\mathcal{H}_4} + \|h \otimes h\|_{\mathcal{H}_4} \\
&\leq \|s\|_{\mathcal{H}_3} + 2\|m\|_{\mathcal{H}_2}\|h\|_{\mathcal{H}_1} + \|h \otimes h\|_{\mathcal{H}_1 \otimes \mathcal{H}_1} \\
&= \|s\|_{\mathcal{H}_3} + 2\|m\|_{\mathcal{H}_2}\|h\|_{\mathcal{H}_1} + \|h\|_{\mathcal{H}_1}^2 \\
&\leq K_h^2 + 2K_h K_m + K_s
\end{aligned}$$

where the second inequality holds by Lemma B.5. Therefore, let $\beta = 1/(K_h^2 + 2K_h K_m + K_s)$ we have $\|\beta f\|_{\mathcal{H}_4} = \beta \|f\|_{\mathcal{H}_4} \leq 1$. Then

$$\begin{aligned}
&\left| \widehat{R}_T(h) - \widehat{R}_S(h) \right| \\
&= \left| \int_{\mathcal{X}} f(\mathbf{x}) dP_T(\mathbf{x}) - \int_{\mathcal{X}} f(\mathbf{x}) dP_S(\mathbf{x}) \right| \\
&= (K_h^2 + 2K_h K_m + K_s) \left| \int_{\mathcal{X}} \beta f(\mathbf{x}) dP_T(\mathbf{x}) - \int_{\mathcal{X}} \beta f(\mathbf{x}) dP_S(\mathbf{x}) \right| \\
&\leq (K_h^2 + 2K_h K_m + K_s) \sup_{\|f\|_{\mathcal{H}_4} \leq 1} \left| \int_{\mathcal{X}} f(\mathbf{x}) dP_T(\mathbf{x}) - \int_{\mathcal{X}} f(\mathbf{x}) dP_S(\mathbf{x}) \right| \\
&= (K_h^2 + 2K_h K_m + K_s) \, \mathrm{MMD}_{k_4}(\mathbf{X}_S, \mathbf{X}_T)
\end{aligned}$$

where $P_T$ denotes the empirical distribution constructed by $\mathbf{X}_T$, so does $P_S$. Recall Lemma B.3, we have Theorem 5.4. $\qquad \square$

*Proof for Theorem 5.6.* Following the notations in Appendix A, we further define

$$\mathbf{w}_* = \mathbf{1}/n, C_\alpha^2 = \mathrm{MMD}^2_{k,\alpha,\mathbf{X}_n}(\mathbf{w}_*) = (1-\alpha)^2 \overline{K} \tag{11}$$

$$\widehat{\mathbf{w}} = \underset{\mathbf{1}^\top \mathbf{w}=1}{\arg\min} \, \mathrm{MMD}^2_{k,\alpha,\mathbf{X}_n}(\mathbf{w}) = \alpha \left( \mathbf{K}^{-1} - \frac{\mathbf{K}^{-1}\mathbf{1}\mathbf{1}^\top\mathbf{K}^{-1}}{\mathbf{1}^\top\mathbf{K}^{-1}\mathbf{1}} \right) \mathbf{p} + \frac{\mathbf{K}^{-1}\mathbf{1}}{\mathbf{1}^\top\mathbf{K}^{-1}\mathbf{1}}$$

Let $\mathbf{p}_\alpha = \alpha\mathbf{p}$, we have $(\mathbf{p}_\alpha - \mathbf{K}\widehat{\mathbf{w}}) \propto \mathbf{1}$. Define

$$\Delta_\alpha(\mathbf{w}) := \text{MMD}^2_{k,\alpha,\mathbf{X}_n}(\mathbf{w}) - C^2_\alpha = \widehat{g}(\mathbf{w}) - \widehat{g}(\mathbf{w}_*)$$

where $\widehat{g}(\mathbf{w}) = (\mathbf{w} - \widehat{\mathbf{w}})^\top \mathbf{K}(\mathbf{w} - \widehat{\mathbf{w}})$. The related details for proving the equality are omitted, since they are completely given by the proof of alternative expression of MMD in Pronzato [32]. By the convexity of $\widehat{g}(\cdot)$, for $j = \arg\min_{i\in[n]\setminus\mathcal{I}^*_p} f_{\mathcal{I}^*_p}(\mathbf{x}_i)$,

$$\widehat{g}(\mathbf{w}_*) \geq \widehat{g}(\mathbf{w}_p) + 2(\mathbf{w}_* - \mathbf{w}_p)^\top \mathbf{K}(\mathbf{w}_p - \widehat{\mathbf{w}}) \geq \widehat{g}(\mathbf{w}_p) + 2\min_{j\in[n]\setminus\mathcal{I}^*_p}(\mathbf{e}_j - \mathbf{w}_p)^\top \mathbf{K}(\mathbf{w}_p - \widehat{\mathbf{w}})$$

where the second inequality holds with the assumption in Theorem 5.6

$$
\begin{aligned}
(\mathbf{w}_* - \mathbf{e}_j)^\top \mathbf{K}(\mathbf{w}_p - \widehat{\mathbf{w}}) &= (\mathbf{w}_* - \mathbf{e}_j)^\top (\mathbf{K}\mathbf{w}_p - \mathbf{p}_\alpha) \\
&= \frac{\sum_{i=1}^n f_{\mathcal{I}^*_p}(\mathbf{x}_i)}{n} - f_{\mathcal{I}^*_p}(\mathbf{x}_{j_{p+1}}) \geq \frac{\sum_{i=1}^n f_{\mathcal{I}^*_p}(\mathbf{x}_i)}{n} - f_{\mathcal{I}^*_p}(\mathbf{x}_j) \geq 0
\end{aligned}
$$

therefore, we have for $B = 2K$,

$$
\begin{aligned}
&\Delta_\alpha(\mathbf{w}_{p+1}) \\
=&\widehat{g}(\mathbf{w}_p) - \widehat{g}(\mathbf{w}_*) + \frac{2}{p+1}(\mathbf{e}_j - \mathbf{w}_p)^\top \mathbf{K}(\mathbf{w}_p - \widehat{\mathbf{w}}) + \frac{1}{(p+1)^2}(\mathbf{e}_j - \mathbf{w}_p)^\top \mathbf{K}(\mathbf{e}_j - \mathbf{w}_p) \\
=&\frac{p}{p+1}(\widehat{g}(\mathbf{w}_p) - \widehat{g}(\mathbf{w}_*)) + \frac{1}{(p+1)^2}B = \frac{p}{p+1}\Delta_\alpha(\mathbf{w}_p) + \frac{1}{(p+1)^2}B
\end{aligned}
\tag{12}
$$

where $\mathbf{w}_{p+1} = p\mathbf{w}_p/(p+1) + \mathbf{e}_j/(p+1)$, and obviously $B$ upper bounds $(\mathbf{e}_j - \mathbf{w}_p)^\top \mathbf{K}(\mathbf{e}_j - \mathbf{w}_p)$. Since $\alpha \leq 1$, it holds from Lemma B.6 that

$$\Delta_\alpha(\mathbf{w}_1) \leq \text{MMD}^2_{k,\alpha,\mathbf{X}_n}(\mathbf{w}_1) \leq (1+\alpha^2)K \leq B$$

therefore by Lemma B.1, we have

$$\text{MMD}^2_{k,\alpha}(\mathbf{X}_{\mathcal{I}^*_m}, \mathbf{X}_n) = \text{MMD}^2_{k,\alpha,\mathbf{X}_n}(\mathbf{w}_p) \leq C^2_\alpha + B\frac{2 + \log m}{m+1}$$

$\square$

# D   Additional Experimental Details and Results

## D.1   Supplementary Numerical Experiments on GKHR

Consider the fact that GKH is a convergent algorithm (Lemma B.2) and the finite-sample-size error bound (10) holds without any assumption on the data, we conduct some numerical experiments to empirically compare GKHR with GKH on datasets generated by four different distributions on $\mathbb{R}^2$.

Firstly, we define four distributions on $\mathbb{R}^2$:

1. Gaussian mixture model 1 which consists of four Gaussian distributions $G_1, G_2, G_3, G_4$ with mixture weights $[0.95, 0.01, 0.02, 0.02]$,

2. Gaussian mixture model 2 which consists of four Gaussian distributions $G_1, G_2, G_3, G_4$ with mixture weights $[0.3, 0.2, 0.15, 0.35]$,

3. Uniform distribution 1 which consists of a uniform distribution defined in a circle with radius $0.5$, and a uniform distribution defined in a annulus with inner radius $4$ and outer radius $6$,

4. Uniform distribution 2 defined on $[-10, 10]^2$.

where

$$G_1 = \mathcal{N}\left(\begin{bmatrix}1\\2\end{bmatrix}, \begin{bmatrix}2 & 0\\0 & 5\end{bmatrix}\right), G_2 = \mathcal{N}\left(\begin{bmatrix}-3\\-5\end{bmatrix}, \begin{bmatrix}1 & 0\\0 & 2\end{bmatrix}\right)$$

$$G_3 = \mathcal{N}\left(\begin{bmatrix}-5\\4\end{bmatrix}, \begin{bmatrix}8 & 0\\0 & 6\end{bmatrix}\right), G_4 = \mathcal{N}\left(\begin{bmatrix}15\\10\end{bmatrix}, \begin{bmatrix}4 & 0\\0 & 9\end{bmatrix}\right)$$

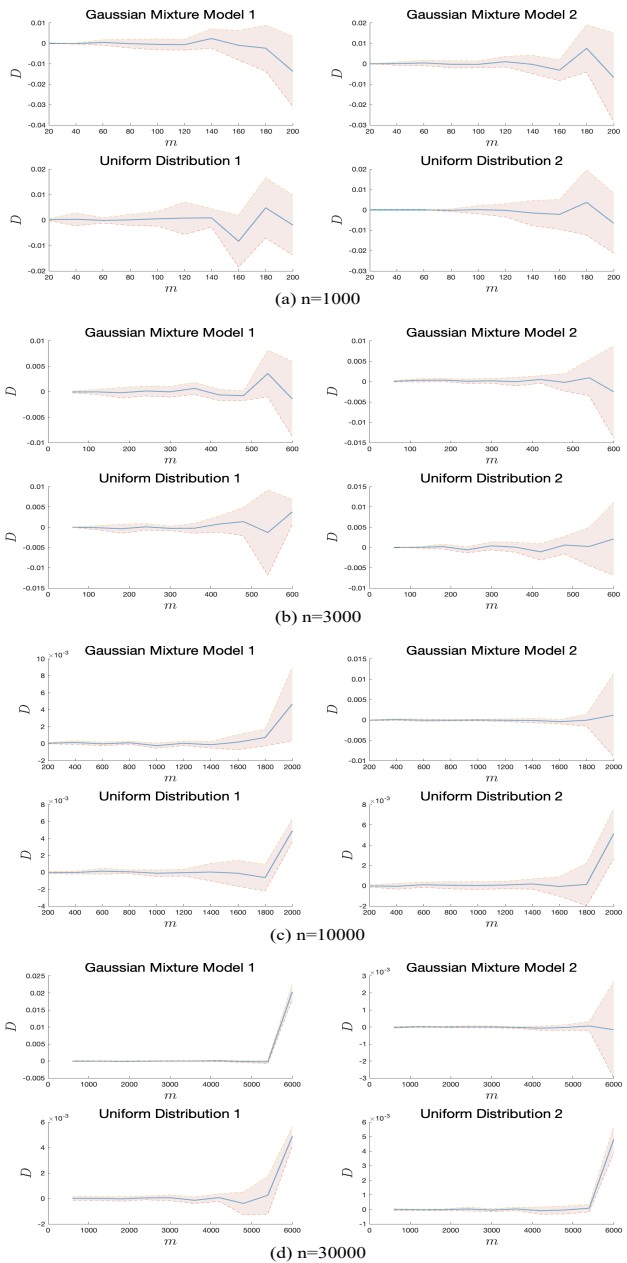

Figure 2: The performance comparison between GKHR and GKH with different $m, n$ over ten independent runs. The blue line is the mean value of $D$, the red dotted line over (under) the blue line is the mean value of $D$ plus (minus) its standard deviation, and the pink area is the area between the upper and lower red dotted lines.

To consistently evaluate the performance gap between GKHR and GKH at the same order of magnitude, we propose the following criterion

$$D = \frac{D_1 - D_2}{D_1 + D_2}$$

where $D_1 = \text{MMD}^2_{k,\alpha}(\mathbf{X}^{(1)}_{\mathcal{I}^*_m}, \mathbf{X}_n), D_2 = \text{MMD}^2_{k,\alpha}(\mathbf{X}^{(2)}_{\mathcal{I}^*_m}, \mathbf{X}_n), \mathbf{X}^{(1)}_{\mathcal{I}_m}$ is the selected samples from GKHR and $\mathbf{X}^{(2)}_{\mathcal{I}_m}$ is the selected samples from GKH. Positive value of $D$ implies that GKH outperforms GKHR, and negative values of $D$ implies that GKHR outperforms GKH. Large absolute value of $D$ shows large performance gap.

The experiments are conducted as follows. We generate 1000,3000,10000,30000 random samples from the four distributions separately, then use GKHR and GKH for sample selection under the low-budget setting, *i.e.,* $m/n \leq 0.2$. The $\alpha$ is set by $m/n$. We report the results over ten independent runs in Figure 2, which shows that although the performance gap tends to grow as $m$ grows, when $m$ is relatively small, the performance of GKHR is similar to that of GKH. Therefore, under the low-budget setting, GKHR and GKH have similar performance on minimizing $\alpha$-MMD over various type of distributions, which convinces us that GKHR could work well in the sample selection task.

## D.2 Datasets

For experiments, we choose five common datasets: CIFAR-10/100, SVHN, STL-10 and ImageNet. CIFAR-10 and CIFAR-100 contain 60,000 images with 10 and 100 categories, respectively, among which 50,000 images are for training, and 10,000 images are for testing; SVHN contains 73,257 images for training and 26,032 images for testing; STL-10 contains 5,000 images for training, 8,000 images for testing and 100,000 unlabeled images as extra training data. ImageNet spans 1,000 object classes and contains 1,281,167 training and 100,000 test images. The training sets of the above datasets are considered as the unlabeled dataset for sample selection.

## D.3 Visualization of Selected Samples

To offer a more intuitive comparison between various sampling methods, we visualized samples chosen by stratified, random, $k$-Means, USL, ActiveFT and RDSS (ours). We generate 5000 samples from a Gaussian mixture model defined on $\mathbb{R}^2$ with 10 components and uniform mixture weights. One hundred samples are selected from the entire dataset using different sampling methods. The visualisation results in Figure 3 indicate that our selected samples distribute more similarly with the entire dataset than other counterparts.

## D.4 Computational Complexity and Running Time

We compute the time complexity of various sampling methods and recorded the time required to select 400 samples on the CIFAR-100 dataset for each method. The results are presented in Table 5, where $m$ represents the annotation budget, $n$ denotes the total number of samples, and $T$ indicates the number of iterations. The sampling time was obtained by averaging the duration of three independent runs of the sampling code on an idle server without any workload. As illustrated by the results, the sampling efficiency of our method surpasses that of all other methods except for random and stratified sampling. This discrepancy is likely because the execution time of other algorithms is affected by the number of iterations $T$.

Table 5: Efficiency comparison with other sampling methods.

| Method | Time complexity | Time (s) |
|---|---|---|
| Random | $O(n)$ | $\approx 0$ |
| Stratified | $O(n)$ | $\approx 0$ |
| $k$-means | $O(mnT)$ | 579.97 |
| USL | $O(mnT)$ | 257.68 |
| ActiveFT | $O(mnT)$ | 224.35 |
| RDSS (Ours) | $O(mn)$ | 132.77 |

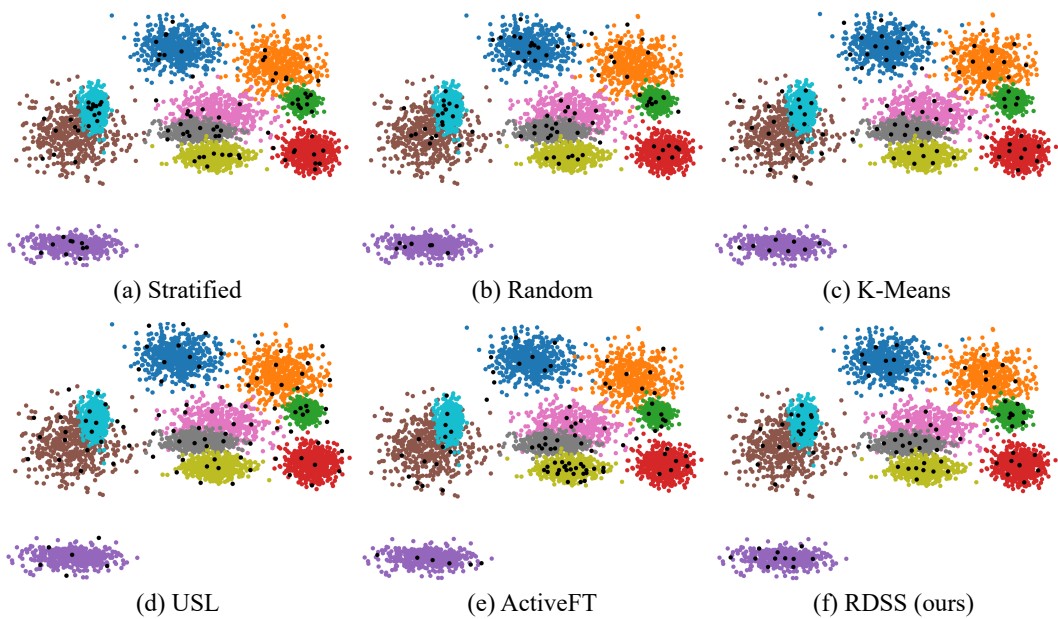

Figure 3: Visualization of selected samples using different sampling methods. Points of different colours represent samples from different classes, while black points indicate the selected samples.

## D.5 Implementation Details of Supervised Learning Experiments

We use ResNet-18 [16] as the classification model for all AL approaches and our method. Specifically, We train the models for 300 epochs using SGD optimizer (initial learning rate=0.1, weight decay=$5e-4$, momentum=0.9) with batch size 128. Finally, we evaluate the performance with the Top-1 classification accuracy metric on the test set.

## D.6 Direct Comparison with AL/SSAL

The comparative results with AL/SSAL approaches are shown in Figure 4 and Figure 5, respectively. The specific values corresponding to the comparative results in the above two figures are shown in Table 6. And the above results are from [8], [12] and [17].

Table 6: Comparative results with AL/SSAL approaches.

| Dataset | CIFAR-10 | | | | | | | | | CIFAR-100 | | | | |
|---|---|---|---|---|---|---|---|---|---|---|---|---|---|---|
| Budget | 40 | 250 | 500 | 1000 | 2000 | 4000 | 5000 | 7500 | 10000 | 400 | 2500 | 5000 | 7500 | 10000 |
| *Active Learning (AL)* | | | | | | | | | | | | | | |
| CoreSet [36] | - | - | - | - | - | - | 80.56 | 85.46 | 87.56 | - | - | 37.36 | 47.17 | 53.06 |
| VAAL [39] | - | - | - | - | - | - | 81.02 | 86.82 | 88.97 | - | - | 38.46 | 47.02 | 53.99 |
| LearnLoss [60] | - | - | - | - | - | - | 81.74 | 85.49 | 87.06 | - | - | 36.12 | 47.81 | 54.02 |
| MCDAL [8] | - | - | - | - | - | - | 81.01 | 87.24 | 89.40 | - | - | 38.90 | 49.34 | 54.14 |
| *Semi-Supervised Active Learning (SSAL)* | | | | | | | | | | | | | | |
| CoreSetSSL [36] | - | - | 90.94 | 92.34 | 93.30 | 94.02 | - | - | - | - | - | 63.14 | 66.29 | 68.63 |
| CBSSAL [12] | - | - | 91.84 | 92.93 | 93.78 | 94.55 | - | - | - | - | - | 63.73 | 67.14 | 69.34 |
| TOD-Semi [17] | - | - | - | - | - | - | 79.54 | 87.82 | 90.3 | - | - | 36.97 | 52.87 | 58.64 |
| *Semi-Supervised Learning (SSL) with RDSS* | | | | | | | | | | | | | | |
| FlexMatch+RDSS (Ours) | 94.69 | 95.21 | - | - | - | 95.71 | - | - | - | 48.12 | 67.27 | - | - | 73.21 |
| FreeMatch+RDSS (Ours) | 95.05 | 95.50 | - | - | - | 95.98 | - | - | - | 48.41 | 67.40 | - | - | 73.13 |

According to the results, we have several observations: (1) AL approaches often necessitate significantly larger labelling budgets, exceeding RDSS by 125 or more on CIFAR-10. This is primarily because AL paradigms are solely dependent on labelled samples not only for classification but also for feature learning. (2) SSAL and our methods leverage unlabeled samples, surpassing traditional AL approaches. However, this may not directly reflect the advantages of RDSS, as such performance enhancements could be inherently attributed to the SSL paradigm itself. Nonetheless, these experimental outcomes offer insightful implications: SSL may represent a more promising paradigm under scenarios with limited annotation budgets.

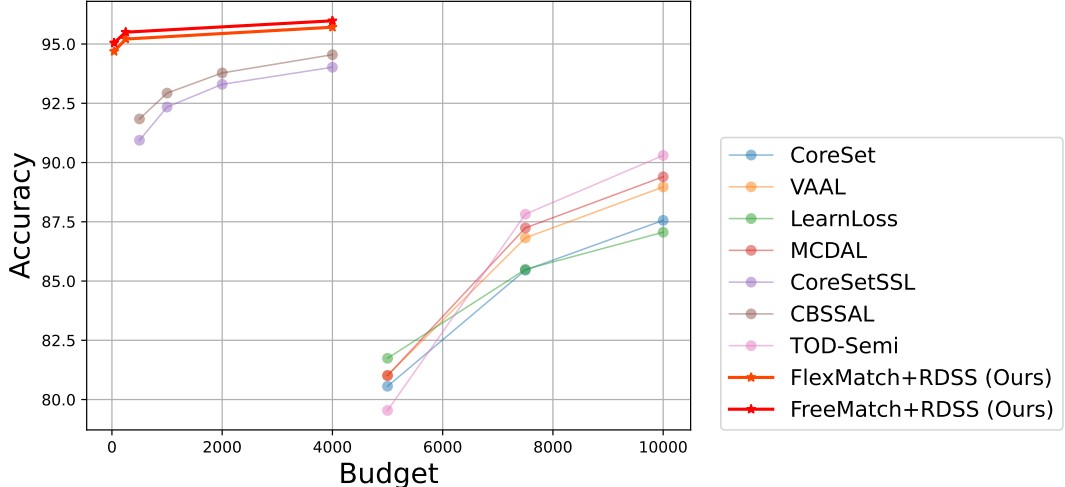

Figure 4: Comparison with AL/SSAL approaches on CIFAR-10.

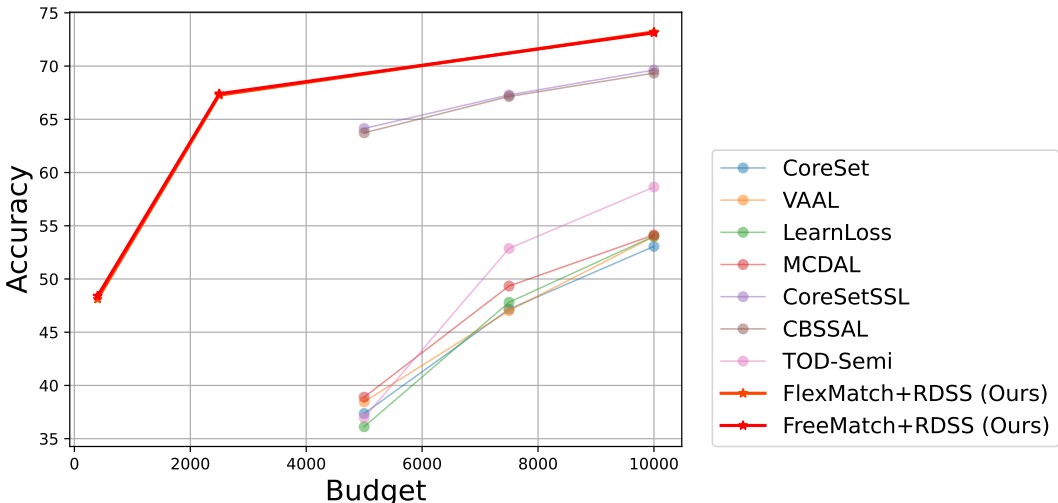

Figure 5: Comparison with AL/SSAL approaches on CIFAR-100.

# E    Limitation

The choice of $\alpha$ depends on the number of full unlabeled data points, independent of the information on the shape of data distribution. This may lead to a loss of effectiveness of RDSS on those datasets with complicated distribution structures. However, it outperforms fixed-ratio approaches on the datasets under different budget settings.

# F    Potential Societal Impact

**Positive societal impact.** Our method ensures the representativeness and diversity of the selected samples and significantly improves the performance of SSL methods, especially under low-budget settings. This reduces the cost and time of data annotation and is particularly beneficial for resource-constrained research and development environments, such as medical image analysis.

**Negative societal impact.** When selecting representative data for analysis and annotation, the processing of sensitive data may be involved, increasing the risk of data leakage, especially in sensitive fields such as medical care and finance. It is worth noting that most algorithms applied in these sensitive areas are subject to this risk.

