# OpenReview forum: "Enhancing Semi-Supervised Learning via Representative and Diverse Sample Selection"
_NeurIPS.cc/2024/Conference — NeurIPS 2024 poster_

### Official Review · Reviewer_CxrG · 2024-07-02

**Soundness:** 2
**Presentation:** 3
**Contribution:** 2
**Rating:** 4
**Confidence:** 5

**Summary:**

The paper suggests a new sampling method for the labeled set of semi-supervised learning. This sampling method, termed RDSS, selects a set of examples that is both representative of the data, and diverse. The paper shows that using such a sampling function improves both freematch and flexmatch, and compares it against other sampling methods, and methods from AL and SSAL.

**Strengths:**

The idea of the paper is good, and is well supported by theory. The experimental setup does convince me that the suggested method is better than random sampling when picking the labeled set of SSL. However, a better comparison to previous works is required, see the Weaknesses section.

Clarity: The paper is clearly written, the idea is well presented and intuitive, and the paper is easy to read and follow.

**Weaknesses:**

Some of the claims made by paper already appeared in previous art. Specifically, [1] showed that "traditional" AL methods do not pick bad labeled sets for SSL when compared to random sampling. [2] showed that when the labeled set is particularly small, instead of traditional AL techniques, one should focus on labeling examples that are more typical and diverse, showing that such a method can drastically improve both AL and sampling techniques for SSL. [3] presented sampling strategy, showing that picking examples that are representative and diverse examples for the labeled set of  SSL improves it by a big margin in low-budget scenarios.

The proposed manuscript does not reference or compare to any of these works. This affects both the novelty, significance and quality of the proposed method: the novelty is somewhat more limited, as many of the ideas overlap with existing works. The significance of this work is impacted, as while the problem at hand is important, it is unclear if the presented ideas pose significant advancement over the existing methods, and the quality is diminished, as a lot of comparisons are missing in the experimental setup.

Specifically, any low-budget strategy could be potentially applied to SSL as well, so those methods should be compared against as well. See for example [4], [5].

Additionally, the vice-versa argument should also hold -- if AL methods can be applied in this case, this method can be used as a method for picking labeled examples for active learning purposes and should be tested as such, as the literature in AL is much broader than the literature of picking the labeled set in SSL, which can provide a much wider context for the given work.

In addition, the framing of the paper is a bit unclear to me. I think the paper could benefit from explaining use cases in which one has the option to pick in advance the labeled set for SSL, which is not already covered by AL use cases.

-------

[1] Mittal, Sudhanshu, et al. Parting with illusions about deep active learning. (2019).

[2] Hacohen, Guy et al. Active learning on a budget: Opposite strategies suit high and low budgets. (2022).

[3] Yehuda, Ofer, et al. "Active learning through a covering lens." (2022).

[4] Mahmood, Rafid, et al. "Low budget active learning via wasserstein distance: An integer programming approach." (2021).

[5] Wen, Ziting, et al. "NTKCPL: Active Learning on Top of Self-Supervised Model by Estimating True Coverage." (2023).

**Questions:**

Can you please elaborate on how the proposed method differs from the idea suggested in [2]?

How is the problem of selecting the labeled set of SSL different from the problem setting of active learning?

**Limitations:**

not relevant

---

> ### Author Rebuttal · Authors · 2024-08-07
>
> Thank you very much for your careful review of our work. We hope our responses can address all your concerns.
>
> **W1. Some of the claims made by paper already appeared in previous art [1-3]. The proposed manuscript does not reference or compare to any of these works.**
>
> **Response:** Thank you for your comment. It is indeed correct that certain aspects of our methodology and initial hypotheses draw from established findings in the literature. The inclusion of these elements is essential for building upon a solid foundation of existing research. However, our work makes significant advancements beyond these foundational elements.
>
> Specifically, even if our method is based on the perspective of representative and diverse sampling that previous studies have proposed, as we said to Reviewer Qb3b, multiple combinations of representativeness and diversity criteria are applicable to defining sample selection optimization objectives, but one should also provide an efficient algorithm and some theoretical guarantees derive an effective methodology that has better generalizability. In our paper, we artfully use kernel methods and the properties of RKHS to derive a simple but effective approach to model the sample selection problem that fulfils the aforementioned requirements for a generalizable method. In the experimental part, we have compared our method with more recent methods (e.g., USL [6] and ActiveFT [7]) and have achieved SOTA performance (refer to Table 1).
>
> **W2. Any low-budget strategy could be potentially applied to SSL as well, so those methods should be compared against as well. See for example [4,5].**
>
> **Response:** We follow your suggestions and compare our proposed RDSS with [4] and [5] by integrating them into FlexMatch and FreeMatch on CIFAR-10 with 40 labels. The experimental results show that RDSS outperforms the two methods.
>
> | Methods         | FlexMatch | FreeMatch |
> | --------------  | --------- | --------- |
> | Wasserstein [4] | 91.64     | 92.72     |
> | NTKCPL [5]      | 87.47     | 91.30     |
> | RDSS (Ours)     | **94.69** | **95.05** |
> |  |  |  |
>
> **W3. If AL methods can be applied in this case, this method can be used as a method for picking labeled examples for active learning purposes and should be tested as such.**
>
> **Response:** Thank you for your suggestions. In fact, we have already conducted comparisons with several SOTA AL methods, as demonstrated in Table 2 of the original manuscript. To ensure a more comprehensive evaluation, we have further supplemented our study with additional experiments comparing our approach against various AL methods. The results of these extended experiments can be found in our response to Reviewer Qb3b's Weakness 3.
>
> **W4. The framing of the paper is a bit unclear to me. I think the paper could benefit from explaining use cases in which one has the option to pick in advance the labeled set for SSL, which is not already covered by AL use cases.**
>
> **Response:** Thank you for your insightful feedback regarding the framing of the paper. We understand that the distinction between the use cases of SSL and AL may not have been sufficiently clear, and we appreciate the opportunity to clarify this aspect. In this work, we focus on scenarios where one can select a labelled set for SSL in advance, a situation that is distinct from traditional AL use cases. A particularly relevant example is in the context of medical trials. In such trials, labelling data (e.g., annotating medical images or patient records) often involves significant time and resources, typically requiring expert review. If the labelled data in later rounds of the trial must be determined based on the outcomes of previous rounds, as in a typical AL setup, the entire project timeline could be significantly extended due to the sequential nature of labelling and analysis. We will revise the manuscript to explicitly differentiate these use cases.
>
> **Q1. Can you please elaborate on how the proposed method differs from the idea suggested in [2]?**
>
> **Response:** Thank you for your question. Our standpoints are all from the idea of representative and diverse sampling, but each method has its own criterion for representativeness and diversity. This situation also happens when comparing our method to USL [6] and ActiveFT [7]. However, among all these works, our method exhibits not only a theoretical guarantee for generalization error but also a fast and effective optimization algorithm for practical sample selection tasks. Moreover, our method can be implemented in some research problems in the literature of statistical subsampling (e.g. [8]), which shows its potential in interdisciplinary scenarios or problems.
>
> **Q2. How is the problem of selecting the labeled set of SSL different from the problem setting of active learning?**
>
> **Response:** We understand that this is a critical issue. In SSL, the selection of the labelled set is typically predetermined or randomly sampled from an available pool of labelled data. In contrast, AL is an iterative process where the model actively queries an oracle (e.g., a human annotator) to label new samples that are expected to be the most informative for improving the model.
>
> References:
> [1] Parting with illusions about deep active learning. arXiv preprint, 2019.
> [2] Active learning on a budget: Opposite strategies suit high and low budgets. ICML, 2022.
> [3] Active learning through a covering lens. NeurIPS, 2022.
> [4] Low budget active learning via wasserstein distance: An integer programming approach. ICLR, 2022.
> [5] NTKCPL: Active Learning on Top of Self-Supervised Model by Estimating True Coverage. arXiv preprint, 2023.
> [6] Unsupervised selective labeling for more effective semi-supervised learning. ECCV, 2022.
> [7] Active Finetuning: Exploiting Annotation Budget in the Pretraining-Finetuning Paradigm. CVPR, 2023.
> [8] Optimal Subsampling via Predictive Inference. Journal of the American Statistical Association, 2023.

---

> > ### Comment · Reviewer_CxrG · 2024-08-11
> > **answer to the rebuttal**
> >
> > I appreciate the authors' detailed rebuttal and the efforts made to address the concerns raised.
> >
> > However, my major concern regarding the comparison to other works in the field remains. While I acknowledge the distinction between active learning (AL) and semi-supervised learning (SSL), I still believe that when the labeled set in SSL is predetermined, it can be viewed as a specific instance of AL—where the initial budget is zero, and the labeled set chosen for SSL serves as the active set. Given the extensive body of work addressing these settings in AL, particularly in low-budget AL scenarios, I believe the paper should include a more thorough comparison to such works. This would better position the paper within the current literature and clarify how it advances the state of the art, if at all.
> >
> > Some of these concerns have been alleviated by the authors' inclusion of an initial comparison to low-budget AL methods in the rebuttal. However, I still find the current framing of the paper somewhat misleading. A peer-reviewed revision would be necessary to incorporate the required changes effectively.
> >
> > In light of the partial resolution of my concerns, I am adjusting my score to 4. Nevertheless, I remain inclined to recommend rejection of the current revision.

---

> > > ### Author Response · Authors · 2024-08-13
> > > **Comparison with sampling methods used in low-budget AL scenarios**
> > >
> > > Thank you very much for taking the time to review our rebuttal. We acknowledge your concern regarding the comparison with other sampling methods used in low-budget AL scenarios [1-3]. While these methods can indeed achieve satisfactory performance in low-budget AL scenarios, they each have inherent limitations that prevent them from effectively adapting to SSL. Specifically, [1] and [3] involve a clustering step before sampling, which makes the sampling results heavily dependent on the quality of the clustering. Consequently, these methods require task-specific adjustments to the clustering algorithm, limiting their general applicability and making them unsuitable for our scenario. In contrast, [2] relies on multiple iterative rounds to enhance the model’s performance for more accurate selection of labelled samples. This iterative nature poses challenges when these methods are employed to determine labelled samples in a single pass, leading to suboptimal outcomes.
> > >
> > > We have incorporated the above methods into FlexMatch and FreeMatch under four low-budget settings: CIFAR-10 with 40 labels, CIFAR-100 with 400 labels, SVHN with 250 labels, and STL-10 with 40 labels, and present the results in the table below. As observed, our method consistently outperforms across all settings, underscoring the superiority of our approach. Furthermore, in the theoretical section, we provide rigorous guarantees to demonstrate that our method is not only efficient but also generalizable.
> > >
> > > | Dataset       | CIFAR-10 (40) | CIFAR-100 (400) | SVHN (250) | STL-10 (40) |
> > > | ------------- | ---           | ---             | ---        | ---         |
> > > | *Applied to FlexMatch* |      |                 |            |             |
> > > | TypiClust     | 91.58         | 46.57           | 90.36      | 74.44       |
> > > | Wasserstein   | 91.64         | 46.76           | 90.22      | 73.45       |
> > > | NTKCPL        | 87.47         | 44.48           | 91.13      | 73.69       |
> > > | RDSS (Ours)   | **94.69**     | **48.12**       | **91.70**  | **77.96**   |
> > > | *Applied to FreeMatch* |      |                 |            |             |
> > > | TypiClust     | 92.38         | 47.26           | 93.09      | 77.30       |
> > > | Wasserstein   | 92.72         | 46.53           | 92.12      | 74.18       |
> > > | NTKCPL        | 91.30         | 46.17           | 91.73      | 78.06       |
> > > | RDSS (Ours)   | **95.05**     | **48.41**       | **94.54**  | **81.90**   |
> > > |  |  |  |  |  |
> > >
> > > [1] Active learning on a budget: Opposite strategies suit high and low budgets. ICML, 2022.
> > > [2] Low budget active learning via wasserstein distance: An integer programming approach. ICLR, 2022.
> > > [3] NTKCPL: Active Learning on Top of Self-Supervised Model by Estimating True Coverage. arXiv preprint, 2023.

---

### Official Review · Reviewer_sn3m · 2024-07-11

**Soundness:** 3
**Presentation:** 3
**Contribution:** 3
**Rating:** 5
**Confidence:** 3

**Summary:**

This paper proposes a Representative and Diverse Sample Selection approach (RDSS) that utilizes a modified Frank-Wolfe algorithm to minimize a novel α-Maximum Mean Discrepancy (α-MMD) criterion, aiming to select a representative and diverse subset from unlabeled data for annotation. Experimental results demonstrate that RDSS consistently improves the performance of several popular semi-supervised learning frameworks and outperforms state-of-the-art sample selection methods used in Active Learning (AL) and Semi-Supervised Active Learning (SSAL), even under constrained annotation budgets.

**Strengths:**

1.This paper is in Well-written, logically organized, and smoothly expressed.
2. The presented results demonstrate the effectiveness of the proposed approach.

**Weaknesses:**

1. The author conducted tests on two baseline methods(FlexMatch [58] and Freematch [50]), but neither of them represents the current state-of-the-art.
2. Some details of the experiments are unclear, such as in Table 3.

**Questions:**

1. The definition and usage of variable X in the article are inconsistent.
2. Is Y in the section starting from line 141 representing the point as X? If so, I suggest using notations like Xi, Xj for clarity.
3. In Chapter 6, the determination of the kernel and parameters seems arbitrary. Could you provide some proofs, theories, or experiments to justify them?
4. The SOTA models selected by the author in the experiments are somewhat outdated. It is recommended to include some more updated methods.
5. In the experiment section, the author mentions the limitations of stratified sampling. What do these limitations refer to? Why was this method excluded? From the results, it seems that stratified sampling outperforms the proposed method in several settings.
6. In Table 2, what sampling method did the other comparison methods adopt?
7. What dataset does Table 3 represent? What is the data distribution?
8. The author should objectively evaluate their method, including its limitations.

**Limitations:**

as Weaknesses

---

> ### Author Rebuttal · Authors · 2024-08-07
>
> Thank you very much for your insightful comments. Since Weakness 1 and Question 4, as well as Weakness 2 and Question 7, address the same issue, we have consolidated them accordingly.
>
> **Q1. The definition and usage of variable X in the article are inconsistent.**
>
> **Response:** Thank you for your careful review of our work. In this article, $\mathbf{X}$ is used for describing datasets, $\mathcal{X}$ denotes the feature space, and X is denoted as random variables. Could you please point out the inconsistent part?
>
> **Q2. Is Y in the section starting from line 141 representing the point as X? If so, I suggest using notations like Xi, Xj for clarity.**
>
> **Response:** We understand your concern. From line 141, y and Y are used rigorously to explain kernel tricks and the definition of MMD, which are commonly used in statistics literature. Nevertheless, we will consider your suggestion for the modification.
>
> **Q3. In Chapter 6, the determination of the kernel and parameters seems arbitrary.**
>
> **Response:** Here is the determination of the kernel and parameters.
> *Kernel:* Our determination of kernel is based on the kernel choice in the study of two-sample test by [1]. In fact, according to Remark 1 (line 170), you can see that any positive-definite and characteristic kernel is applicable to our method. Among all kernels, Gaussian kernels are the most popular in the study of machine learning, so we take them as our suggestion.
>
> *Bandwidth parameters:* The choice is also based on [1]. So far there is no theoretical guidance for us to determine the bandwidth parameters of Gaussian kernels in measuring similarity or representativeness. Some methods optimize this parameter in the learning process, but this idea is not applicable in RDSS.
>
> $\alpha$: According to lines 246-250, we derived a finite-sample-error bound for MMD, which measures the representativeness, leading to our suggestion of the range of $\alpha$. In the experiments, the choice of $\alpha=1-1/\sqrt{m}$ significantly outperforms other choices.
>
> **Q4 & W1. The SOTA models selected by the author in the experiments are somewhat outdated.**
>
> **Response:** Thank you for your suggestion. Since the sampling process and the SSL process are relatively independent, comparing the sampling methods across different SSL models may have a limited impact. However, we have still compared different methods under two additional SOTA SSL approaches, i.e., ReFixMatch [2] and SequenceMatch [3]. The experimental results in the table below show that RDSS achieves the highest accuracy.
>
> | Dataset              | CIFAR-10 |     |      | CIFAR-100  |      |      | SVHN |     | STL-10 |     |
> | -------------------- | ---      | --- | ---  | ---        | ---  | ---  | ---  | --- | ---    | --- |
> | Budget               | 40       | 250 | 4000 | 400        | 2500 | 10000| 250  | 1000| 40     | 250 |
> | *Applied to ReFixMatch* |        |     |      |            |      |      |      |     |        |     |
> | ActiveFT    | 75.62 | 93.56 | 95.64 | 26.86 | 56.97 | 71.65 | 96.17 | 96.97 | 62.53 | 87.53 |
> | USL         | 94.34 | 95.12 | 95.77 | 48.23 | 66.61 | 72.33 | 96.16 | 97.17 | 75.38 | 91.24 |
> | RDSS (Ours) | **95.18** | **95.54** | **96.22** | **49.32** | **67.46** | **73.15** | **96.75** | **97.52** | **78.78** | **92.25** |
> | *Applied to SequenceMatch* |        |     |      |            |      |      |      |     |        |     |
> | ActiveFT    | 78.91 | 95.02 | 95.11 | 31.94 | 55.49 | 70.38 | 93.65 | 94.20 | 69.44 | 89.44 |
> | USL         | 94.12 | 95.10 | 95.85 | 49.34 | 67.04 | 73.68 | 95.22 | 96.04 | 76.63 | 90.13 |
> | RDSS (Ours) | **95.33** | **95.26** | **96.17** | **50.76** | **69.96** | **74.83** | **96.86** | **97.91** | **81.41** | **92.86** |
> |  |  |  |  |  |  |  |  |  |  |  |
>
> **Q5. What are the limitations of stratified sampling. Why was this method excluded?**
>
> **Response:** Stratified sampling requires prior knowledge of sample categories, and then random sampling is performed within each category. However, in many real-world scenarios, we do not have prior knowledge of sample categories, making stratified sampling inapplicable. This is its limitation. We have a more comprehensive description in the manuscript (Line 25 and Line 81).
>
> **Q6. In Table 2, what sampling method did the other comparison methods adopt?**
>
> **Response:** We are sorry for overlooking this. The sampling methods employed by the AL approaches in Table 2 are as follows:
> *CoreSet* uses a greedy algorithm known as $k$-Center-Greedy to select a subset from the unlabeled dataset;
> *VAAL* consists of a VAE and an adversarial network. The VAE tries to trick the adversarial network into predicting that all data points are from the labelled pool, the adversarial network learns how to discriminate between dissimilarities in the latent space. The samples predicted as "unlabeled" by the adversarial network are selected for labelling;
> *LearnLoss* designes a loss prediction module for a target network, which predicts target losses of unlabeled samples. The samples with the top-$K$ predicted losses are selected to be labelled;
> *MCDAL* utilizes two auxiliary classification layers to select samples with the largest prediction discrepancy between them as those requiring labelling.
>
> **Q7 & W2. What dataset does Table 3 represent?**
>
> **Response:** Thank you for your reminder. The dataset used in Table 3 is CIFAR-10. We will refine these details in the next version of our manuscript.
>
> **Q8. The author should objectively evaluate their method, including its limitations.**
>
> **Response:** Thank you for your insightful suggestion. We have discussed the limitations of our method in Appendix E.
>
> References:
> [1] A kernel two-sample test. The Journal of Machine Learning Research, 2012.
> [2] Boosting Semi-Supervised Learning by bridging high and low-confidence predictions. ICCV, 2023.
> [3] SequenceMatch: Revisiting the design of weak-strong augmentations for Semi-supervised learning. WACV, 2024.

---

> > ### Comment · Reviewer_sn3m · 2024-08-09
> >
> > Thank you for your rebuttal. I have no quetions.

---

> > > ### Author Response · Authors · 2024-08-09
> > >
> > > Thank you very much for taking the time to review our rebuttal. We appreciate your acknowledgement and are glad to have addressed your concerns. Given the significance of our findings and the rigorous methodology we employed, we believe our work makes a valuable contribution to the SSL field. We would be grateful if you could kindly reconsider your evaluation in light of the clarifications and contributions highlighted in our rebuttal.

---

### Official Review · Reviewer_GzC9 · 2024-07-12

**Soundness:** 3
**Presentation:** 3
**Contribution:** 3
**Rating:** 6
**Confidence:** 2

**Summary:**

This paper proposes a new sample selection method, RDSS, for the SSL task. RDSS considers both the representativeness and diversity of the selected sample and achieves state-of-the-art performance. This is achieved by the proposed α-MMD criterion and an efficient optimization algorithm GKHR.

**Strengths:**

1. RDSS considers both representativeness and diversity of samples, which is a convincing strategy, and the experimental results also demonstrate the effectiveness of this motivation.
2. Sufficient theoretical analysis and experimental comparisons are conducted to demonstrate the effectiveness of the proposed method.

**Weaknesses:**

I would like to see images of the actual selected samples and visualizations of the feature distribution to demonstrate that RDSS indeed balances the representativeness and diversity.

**Questions:**

Please refer to the weakness.

**Limitations:**

Please refer to the weakness.

---

> ### Author Rebuttal · Authors · 2024-08-07
>
> Thank you very much for your insightful comments, which have greatly contributed to improving the quality of our paper. We hope our responses can address all your concerns.
>
> **W1. I would like to see images of the actual selected samples and visualizations of the feature distribution to demonstrate that RDSS indeed balances the representativeness and diversity.**
>
> **Response:** We follow your suggestion and upload a PDF that contains the actual samples selected using different sampling methods in the "global" response. It can be observed that in the samples selected by our method, the variation in the number of samples across different categories is minimal. In contrast, certain methods fail to select representative samples for every category. For instance, $k$-means does not choose any samples from the *horse* category. Additionally, other methods tend to select an excessive number of similar samples within a single category, thereby neglecting diversity. For example, the ActiveFT approach selects too many samples from the *airplane* category.

---

> > ### Comment · Reviewer_GzC9 · 2024-08-12
> >
> > I appreciate the author's rebuttal and the efforts made to address my concern. I have noticed the new visualization results. Additionally, I would like to ask whether the feature distribution and corresponding sample images in Fig 1 are the results of actual experiments or merely conceptual illustrations. If they are just conceptual illustrations, providing the distribution from actual experiments would be more convincing.

---

> > > ### Author Response · Authors · 2024-08-12
> > > **Thank you for your comments!**
> > >
> > > Thank you for your thorough review. We apologize for any confusion caused. The images presented in Figure 1 depict an actual sampling result from our method on the CIFAR-10 dataset, where we set the sample size to 40, rather than serving as a conceptual illustration. More detailed experimental settings can be found in the RDSS.py file within the supplementary materials we have uploaded.

---

> > > > ### Comment · Reviewer_GzC9 · 2024-08-13
> > > >
> > > > Thanks for your explanation. I have no other problems. Good luck!

---

### Official Review · Reviewer_Qb3b · 2024-07-12

**Soundness:** 3
**Presentation:** 3
**Contribution:** 3
**Rating:** 6
**Confidence:** 3

**Summary:**

Choice of the labeled set in the semi supervised learning is critical for the final performance of the model. This problem can also be looked as AL with SSL, or single shot AL with SSL (in other words similar to experimental design). This works provides a way to select the seed set which is representative, as well as diverse. The problem is reduced to minimizing MMD and similarity score of the selected examples. The paper finally proposes a greedy algorithm, and compare the proposed method against various subset selection baselines, and AL.

**Strengths:**

I like the motivation of the problem and a neat theoretical derivation of the objective, and the provided theoretical analysis. Paper was also easy to follow and experiments are compelling.

**Weaknesses:**

- From a purely combinatorial point of view, I think that the final objective is supermodular in nature. Given the vast literature on submodular/supermodular functions, is it not possible to get an algorithm purely from that standpoint? If so, how different would it be from the proposed one?

- Can one derive things such as leverage scores to detect the outlier-ness of a given point (or any other score)? If so, then couldn't one use something such as diversity - outlier score (or add a score that models likelihood) , with diversity such as Facility location function, and optimize the final objective using greedy?

- In experiments I believe one of the strong baselines such as facility location function is missing. Facility Location has a rich history and have been used in several instances in Active Learning ([1, 2, 3, 4]). I believe authors can add a small discussion on FL and add that baseline. Furthermore, other diversity based approaches have also been considered in the past [5]

- Now a days a lot of focus is also for doing finetuning of existing CLIP models [3]. I'd appreciate one experiment on fine-tuning the CLIP models using the proposed method.


References
- [1] Submodularity in machine learning and artificial intelligence
- [2] An Experimental Design Framework for Label-Efficient Supervised Finetuning of Large Language Models
- [3] LabelBench: A Comprehensive Framework for Benchmarking Adaptive Label-Efficient Learning
- [4] Deep Submodular Peripteral networks
- [5] GLISTER: Generalization based Data Subset Selection for Efficient and Robust Learning

**Questions:**

Refer to the weaknesses.

**Limitations:**

Refer to the weaknesses.

---

> ### Author Rebuttal · Authors · 2024-08-07
>
> Thank you very much for your constructive comments, which have definitely helped us enhance the paper and highlight its contributions in a better way.
>
> **W1. Given the vast literature on submodular/supermodular functions, is it not possible to get an algorithm purely from that standpoint? If so, how different would it be from the proposed one?**
>
> **Response:** Thank you for your question. In our opinion, it is possible to derive a sample selection algorithm from submodular/supermodular functions. This idea is inspiring to us, and we will try to explore its realizability. However, in our paper, if we apply submodular/supermodular functions to deal with the optimization problem, the problem can be purely combinatorial and may lose its advantage in low computational complexity, which is achieved by exploring the convexity of the weighed MMD function (see Appendix A.1).
>
> **W2. Can one derive things such as leverage scores to detect the outlier-ness of a given point (or any other score)? If so, then couldn't one use something such as diversity - outlier score (or add a score that models likelihood), with diversity such as Facility location function, and optimize the final objective using greedy?**
>
> **Response:** Thank you for your question. Multiple combinations of representativeness and diversity criteria are applicable to define sample selection optimization objectives by representativeness and diversity. However, to derive an effective methodology, one should also provide an efficient algorithm and some theoretical guarantees, which require a detailed study on the definition of the optimization objectives.
>
> From this perspective, your idea can be effective if your final objective can be modelled by a submodular framework. Intuitively, this can be achievable if we detaily study the properties of facility location function, diversity - outlier score and models likelihood. In our paper, we use kernel methods and RKHS to model the problem and derive GKHR as an efficient algorithm and generalization/finite-sample-error bound as the theoretical guarantee. The artful combination of representativeness (MMD) and diversity (kernel functions) criterions provide us a simple but effective approach to model the sample selection problem, so we don't have to use greedy algorithm to deal with it. Nevertheless, next time, we will try to start from your standpoint to study this problem if possible.
>
> **W3. Comparison with facility location function [1-4] and diversity based approaches [5] is missing.**
>
> **Response:** We follow your suggestions and compare two facility location-based methods from [2], namely the k-center and the conventional Facility Location (FL) method, as well as the GLISTER method [5]. Notably, the k-center method is equivalent to the Coreset method [6], which we have already benchmarked in our original manuscript (refer to line 309, Table 2).
>
> The sampling process in GLISTER is tightly coupled with the downstream classification task, making it impossible to pre-determine all labelled samples, which renders it unsuitable for SSL. Therefore, we conduct comparisons under the AL framework, consistent with the experimental setup described in Appendix D.5 of the original manuscript. The results are presented in the table below, demonstrating that RDSS consistently outperforms the other methods, with particularly significant improvements observed on CIFAR-100 using 7,500 labels. We will incorporate these experimental results into a subsequent version of the manuscript.
>
> | Dataset       | CIFAR-10  |           | CIFAR-100 |           |
> | ------------- | ---       | ---       | ---       | ---       |
> | Budget        | 7500      | 10000     | 7500      | 10000     |
> | k-center      | 85.46     | 87.56     | 47.17     | 53.06     |
> | FL            | 86.03     | 89.21     | 47.87     | 55.45     |
> | GLISTER       | 86.64     | 89.33     | 48.74     | 55.39     |
> | RDSS (Ours)   | **87.18** | **89.77** | **50.13** | **56.04** |
> |  |  |  |  |  |
>
>
> **W4. I'd appreciate one experiment on fine-tuning the CLIP models [3] using the proposed method.**
>
> **Response:** This is a good point. We fine-tune the CLIP model via a selection-via-proxy approach [3]. We compare the RDSS with random and k-center sampling methods on CIFAR-10/100 with 10,000 labelled instances when applied to FlexMatch and FreeMatch. The results are shown in the table below, from which we find that RDSS achieves the highest accuracy, outperforming the other two sampling methods. We will incorporate these experimental results into a subsequent version of the manuscript.
>
> | Dataset       | CIFAR-10  | CIFAR-100 |
> | ------------- | ---       | ---       |
> | *Applied to FlexMatch* |        |     |
> | Random        | 96.46     | 83.37     |
> | k-center      | 96.51     | 85.48     |
> | RDSS (Ours)   | **97.83** | **86.75** |
> | *Applied to FreeMatch* |        |     |
> | Random        | 96.58     | 83.29     |
> | k-center      | 96.75     | 86.14     |
> | RDSS (Ours)   | **98.02** | **86.96** |
> |  |  |  |
>
> References:
> [1] Submodularity in machine learning and artificial intelligence. arXiv preprint, 2022.
> [2] An Experimental Design Framework for Label-Efficient Supervised Finetuning of Large Language Models. arXiv preprint, 2024.
> [3] LabelBench: A Comprehensive Framework for Benchmarking Adaptive Label-Efficient Learning. Journal of Data-centric Machine Learning Research, 2024.
> [4] Deep Submodular Peripteral networks. arXiv preprint, 2024.
> [5] GLISTER: Generalization based Data Subset Selection for Efficient and Robust Learning. AAAI, 2021.
> [6] Active learning for convolutional neural networks: A core-set approach. ICLR, 2018.

---

> > ### Comment · Reviewer_Qb3b · 2024-08-09
> > **Thank you for the rebuttal!**
> >
> > I thank the authors for rebuttal. GLISTER seems to be doing some bi-level optimization, so I am not sure how it is same as solving FL optimization, but what I was hoping for is something similar to done in say [2] (same citations as above).
> >
> >
> > I also thank the authors for adding the results for fine-tuning CLIP model. Ideally I'd appreciate error bars, as well as going beyond CIFAR 10/100 since it is fine-tuning the CLIP model, one can get decent Imagenet performance too. (As in Labelbench)
> >
> >
> > I will retain my score, and hoping the discussion and new results mentioned here in the next version of the manuscript.

---

> > > ### Author Response · Authors · 2024-08-10
> > > **Thank you for your comments!**
> > >
> > > Thank you very much for taking the time to review our rebuttal. We are currently conducting comparative experiments similar to the work in [2] (same citations as above) and fine-tuning the CLIP model on the ImageNet dataset. Due to the time constraints and the large volume of data, we were unable to complete these experiments by the discussion phase. However, we will include these experimental results in the next version of the manuscript.

---

### Author Rebuttal · Authors · 2024-08-07

Here is a visualization of the sampling results for Reviewer GzC9.

---

### Decision · Program_Chairs · 2024-09-25

**Decision:**

Accept (poster)

**Comment:**

This paper studies the sample selection problem for semi-supervised learning and proposes a novel RDSS method by adopting a modified Frank-Wolfe algorithm to minimize a novel criterion α-MMD. During the rebuttal, the authors provide detailed discussion and experimental results corresponding to the questions and weaknesses. The major concerns raised by the reviewers are well addressed. The remaining concern about the difference between the setting of this paper and low-budget AL is properly resolved with the follow-up response. Therefore, this paper meets the standards of NeurIPS and is ready for publication. Please revise this paper according to the reviewers' suggestions in the final version.